

# *HydrothermalFoam* v1.0: a 3-D hydro-thermo-transport model for natural submarine hydrothermal systems

Zhikui Guo[1,2], Lars Rüpke[2], and Chunhui Tao[1,3]

[1]Key Laboratory of Submarine Geosciences, MNR, Second Institute of Oceanography, MNR, Hangzhou 310012, China
[2]GEOMAR Helmholtz Centre for Ocean Research Kiel, Wischhofstr. 1-3, 24159 Kiel, Germany
[3]School of Oceanography, Shanghai Jiao Tong University, 1954 Huashan Rd., Shanghai 200030, China.

**Correspondence:** Lars Rüpke (lruepke@geomar.de) and Chunhui Tao (taochunhuimail@163.com)

**Abstract.** Herein, we introduce *HydrothermalFoam*, a three dimensional hydro-thermo-transport model designed to resolve fluid flow within submarine hydrothermal circulation systems. *HydrothermalFoam* has been developed on the OpenFOAM platform, which is a Finite Volume based C++ toolbox for fluid-dynamic simulations and for developing customized numerical models that provides access to state-of-the-art parallelized solvers and to a wide range of pre- and post-processing tools. We

have implemented a porous media Darcy-flow model with associated boundary conditions designed to facilitate numerical simulations of submarine hydrothermal systems. The current implementation is valid for single-phase fluid states and uses a pure water equation-of-state (IAPWS-97). We here present the model formulation, OpenFOAM implementation details, and a sequence of 1-D, 2-D and 3-D benchmark tests. The source code repository further includes a number of tutorials that can be used as starting points for building specialized hydrothermal flow models. The model is published under the GNU General

Public License v3.0.

## 1  Introduction

High temperature hydrothermal circulation through the ocean floor plays a key role in the exchange of mass and energy between the solid earth and the global ocean (German and Seyfried, 2014; Elderfield and Schultz, 1996). It influences the thermal evolution of young oceanic plates (Stein and Stein, 1994; Theissen-Krah et al., 2016), modulates global ocean biogeochemical

cycles (German et al., 2016; Tagliabue et al., 2010), and is associated with massive sulphide ore deposits that form around vent sites (Hannington et al., 2011). Hydrothermal convection occurs over large spatial and temporal scales. At fast spreading ridges, convection cells may either be confined to the upper extrusive crust above the axial melt lens (Faak et al., 2015; Coumou et al., 2008; Fontaine et al., 2009), or extend all the way down to the crust mantle boundary at approx. 6 km depth (Hasenclever et al., 2014; Dunn et al., 2000; Cathles, 1993). At slow spreading ridges, fluid circulation may extend much deeper (up to 35

20  km) into the ultramafic mantle (Schlindwein and Schmid, 2016), although the maximum extent of the brittle layer remains debated and may be confined to the upper  15 km (Grevemeyer et al., 2019). Such deep-reaching fluid flow can sometimes be channelized along deep detachments at fault-controlled systems such as TAG and Lonqi (Tao et al., 2020; deMartin et al., 2007) and/or may propagate to greater depths via thermal cracking (Olive and Crone, 2018; Lister, 1974). Temperatures can also vary over large ranges with high-temperature systems typically being driven by a magmatic heat source of 1000 °C or more and



porosity/permeability staying open up to 600 - 800 °C (Lister, 1974). Finally, hydrothermal systems can evolve over long times scales of up to 50-100 kyrs (Jamieson et al., 2014) but also respond to shorter events like glacial sealevel changes (Middleton et al., 2016) or magmatic as well as seismic events (Germanovich et al., 2000; Wilcock, 2004; Singh and Lowell, 2015) and even tidal pressure changes (Crone et al., 2011; Barreyre et al., 2018). These spatial and temporal scales in combination with the extreme pressure (P) and temperature (T) conditions (up to 300 MPa and 1000 °C) of submarine hydrothermal systems

make direct and long-term observations challenging and pose a problem for laboratory work (Ingebritsen et al., 2010). Hence, numerical simulations have become indispensable tools for understanding and characterizing fluid flow and for relating seafloor observations to physico-chemical processes at depth.

In the last decades, significant progresses have been made in hydrothermal flow modelling both theoretically and numerically (Lowell, 1991; Ingebritsen et al., 2010). Due to the high complexity of the heterogeneous sub-seafloor, a continuum porous

medium approach based on Darcy's law is typically used, in which the conservation equations are written for control volumes with effective properties such as Darcy velocity, permeability, and porosity. Such approaches have been successfully used to make fundamental progresses in our understanding of the nature and mechanisms of hydrothermal transport, including a thermodynamic explanation of black smoker temperatures (Jupp and Schultz, 2000), the three-dimensional structure of hydrothermal circulation cells at mid-ocean ridges (Coumou et al., 2009; Hasenclever et al., 2014), and phase separation

phenomena as well as salinity variations of hydrothermal fluids (Lewis and Lowell, 2009a, b; Coumou et al., 2009; Weis et al., 2014).

Current numerical simulators of hydrothermal flow can be divided into two families: 1) multi-phase codes that thrive towards resolving saltwater convection and associated phase separation phenomena and 2) single-phase hydrothermal codes that focus on sub-critical low-temperature fluid flow and/or super-critical high-temperature flow of pure water, i.e. codes that only "work"

within single-phase fluid states. Multi-phase saltwater codes are at the forefront of what is currently feasible in numerical simulations as accounting for the complexity of the equation-of-state (EOS) of seawater (Driesner and Heinrich, 2007; Driesner, 2007) in combination with multi-phase transport is a challenge (Ingebritsen et al., 2010). Existing codes of this type include CSMP++, which is capable of treating salt water up to magmatic temperatures on unstructured finite element-finite volume (FEFV) meshes (Weis et al., 2014). The hydrothermal multi-phase version of CSMP++ is currently 2-D and it's a closed-source

project. FISHES is a 2-D open-source academic code, which uses the finite volume method to solve thermohaline convection on structured meshes (Lewis and Lowell, 2009a, b) but has some restrictions on the phase states that can be resolved. Currently there is no 3-D model that can resolve multi-phase saltwater convection but there are some developments efforts on the way. In addition, there are a number of geothermal modeling codes that can handle two-phase behavior that have not (yet) been adapted to handle the complex EOS of saltwater over sufficiently large pressure and temperature ranges. HYDROTHERM (Kipp et al., 2008),

FEHM (Zyvoloski et al., 1997), HT2_NR (Vehling et al., 2018), and TOUGH2 (Pruess et al., 1999) are examples of such codes. The second type of code family refers to somewhat simpler models that circumvent the numerical challenges of multi-phase phenomena by staying in P-T regions, where the simulated fluid is in single-phase. A popular approach is to use a pure water instead of a saltwater EOS at pressures beyond the critical end-point (22 MPa). These models, despite making simplifying assumptions, continue to be widely used in the submarine hydrothermal system community and have been successfully applied





to solve a wide range of problems. Examples include Jupp and Schultz (2000, 2004), who showed that hydrothermal systems operate close to optimal efficiency with their maximum vent temperatures set by the thermodynamic properties of water, studies that revealed the complex 3-D structure of recharge and discharge flow in mid-ocean ridge hydrothermal systems (Hasenclever et al., 2014; Coumou et al., 2008; Fontaine et al., 2014), dedicated case studies for individual vent systems (Tao et al., 2020; Andersen et al., 2015; Lowell et al., 2012), and models exploring tidal forcing of hydrothermal circulation (Crone and Wilcock,

2005; Barreyre et al., 2018). This list is nowhere near complete and there are many more examples. The bottom line is that single phase circulation models continue to be widely used and highly useful "workhorses" in the hydrothermal community. Somewhere in the hopefully not so far future, 2-D and 3-D multi-phase models will be the new standard but for now robust and tested single-phase codes continue to be useful tools for a variety of applications.

   Interestingly, even single-phase models are not that easily accessible to the hydrothermal community. Many research groups

maintain 2-D research codes that resolve hydrothermal flow but single-phase 3-D models continue to be rare. To our knowledge there are basically three single-phase code families that are routinely used in 3-D studies (Coumou et al., 2008; Hasenclever et al., 2014; Fontaine et al., 2014) and none of them is open-source. There are some major open-source initiatives that provide 3-D porous flow simulators or libraries such as Dumux (Flemisch et al., 2011), MRST (Lie, 2019), and OpenGeoSys (Kolditz et al., 2012) that can be used to simulate hydrothermal flow but none of them has been adapted and documented for simulating

submarine hydrothermal systems. In this paper, we present a toolbox, named *HydrothermalFoam*, to simulate 2-D and 3-D hydrothermal circulation in single-phase regime for seafloor hydrothermal systems. The toolbox is build upon the open-source platform OpenFOAM® (Jasak, 1996; Weller et al., 1998), which is not only a widely used simulator for solving Navier-Stokes-type problems but also a general toolbox for solving partial differential equations. OpenFOAM is based on the cell-centroid finite volume method (FVM) and is written in C++. It provides high-level interfaces for field operations and includes a series of

features such as support for flexible meshes (e.g. structured meshes, unstructured mesh, and mixed mesh), utilities of pre- and post-processing, and parallel computing in 2-D and 3-D (Moukalled et al., 2016). Based on this established framework, we present a toolbox to simulate flow in submarine hydrothermal systems. We solve the porous flow problem using a continuum porous medium approach in which the fluid velocity is expressed by Darcy's law and the pressure equation is constructed from Darcy's law and the mass conservation equation. All the partial differential equations are solved implicitly in the framework of

OpenFOAM and the thermal-physical models are developed using a pure water EOS. *HydrothermalFoam* inherits all kinds of basic features of OpenFOAM, including boundary conditions. In addition, we have also customized several special boundary conditions for seafloor hydrothermal system modeling. The purpose of this toolbox is to provide the hydrothermal community with a state-of-the-art yet easy-to-use and well-documented simulator for resolving hydrothermal flow in submarine hydrothermal systems.

The paper is organized as follows. In section 2, we present the mathematical model, its implementation in OpenFOAM, information on initial and boundary conditions, and the thermal physical model selection. In section 3, we describe the different toolbox components, in section 4 the installation options and procedures, and in section 5 the toolbox is validated using several published benchmark tests.





**Table 1.** Definitions and values of variables used in this study

| Symbol | Definition | Value | Unite | Variable name: OpenFOAM class |
|---|---|---|---|---|
| $\boldsymbol{g}$ | Gravitational acceleration vector | 9.81 | $\mathrm{m\,s^{-2}}$ | g: uniformDimensionedVectorField |
| $T$ | Temperature | | K | T: volScalarField |
| $p$ | Pressure | | Pa | p: volScalarField |
| $k$ | Permeability | | $\mathrm{m^2}$ | permeability: volScalarField |
| $\boldsymbol{U}$ | Darcy velocity | | $\mathrm{m\,s^{-1}}$ | U: volVectorField |
| $\Delta T$ | Time step | | s | deltaT_: scalar |
| $C_o$ | Courant number | | | CoNum: scalar |
| $C_{\Delta t}$ | Coefficient for time-step change | | | maxDeltaTFact: scalar |
| $\boldsymbol{q_h}$ | Heat flux | | $\mathrm{W\,m^{-2}}$ | q_: scalarField |
| $\boldsymbol{\phi_g}$ | Gravity related flux | | $\mathrm{kg\,s^{-1}}$ | phig: surfaceScalarField |
| $\boldsymbol{\phi_m}$ | Mass flux | | $\mathrm{kg\,m^{-2}\,s^{-1}}$ | phi: surfaceScalarField |
| $\boldsymbol{n}$ | Normal vector of face | | | |
| *Thermal dynamic properties of fluid (IAPWS 97)* | | | | thermo: hydroThermo |
| $C_{pf}$ | Specific heat of fluid | | $\mathrm{m^2\,s^{-2}\,K^{-1}}$ | Cp: volScalarField |
| $\mu_f$ | Dynamic viscosity of fluid | | Pa s | mu: volScalarField |
| $\rho_f$ | Density of fluid | | $\mathrm{kg\,m^{-3}}$ | rho: volScalarField |
| $\alpha_f$ | Thermal expansivity | | $\mathrm{K^{-1}}$ | alphaP: volScalarField |
| $\beta_f$ | Compressibility | | $\mathrm{Pa^{-1}}$ | betaT: volScalarField |
| *Rock properties* | | | | transportProperties:IOdictionary |
| $\varepsilon$ | Porosity of rock | 0.1 | | porosity: dimensionedScalar |
| $\rho_r$ | Density of rock | 2750 | $\mathrm{kg\,m^{-3}}$ | rho_rock: dimensionedScalar |
| $C_{pr}$ | Specific heat of rock | 880 | $\mathrm{J\,kg^{-1}\,K^{-1}}$ | cp_rock: dimensionedScalar |
| $\lambda_r$ | Thermal conductivity of rock | 1.5 | $\mathrm{W\,m^{-1}\,K^{-1}}$ | kr: dimensionedScalar |

## 2 Model development

### 2.1 Mathematical model of hydrothermal flow

We use a continuum porous media approach and describe laminar flow in hydrothermal circulation systems using Darcy's law, where the Darcy velocity of the fluid is given by

$$\boldsymbol{U} = -\frac{k}{\mu_f}(\nabla p - \rho \boldsymbol{g}) \tag{1}$$

in which $k$ denotes permeability, $\mu_f$ the fluid's dynamic viscosity, $p$ total fluid pressure and $\boldsymbol{g}$ gravitational acceleration. All variables and symbols are listed in Table 1. Considering a compressible fluid in a porous medium with given porosity structure,





the mass balance is expressed by

$$\varepsilon \frac{\partial \rho_f}{\partial t} + \nabla \cdot (\boldsymbol{U} \rho_f) \tag{2}$$

where $\varepsilon$ is the porosity of the rock. The equation for pressure can be derived from Darcy's law (Equation 1) and the continuity equation (2) by assuming that the compressibility of the rock is relative small and hence negligible. Substituting equation (1)

into equation (2) and treating the fluid's density as a function of temperature $T$ and pressure $p$ yields,

$$\varepsilon \rho_f \left( \beta_f \frac{\partial p}{\partial t} - \alpha_f \frac{\partial T}{\partial t} \right) = \nabla \cdot \left( \rho_f \frac{k}{\mu_f} (\nabla p - \rho_f \boldsymbol{g}) \right) \tag{3}$$

with $\alpha_f$ and $\beta_f$ being the fluid's thermal expansivity and compressibility, respectively. Energy conservation of a single-phase fluid can be expressed using a temperature formulation (Hasenclever et al., 2014),

$$(\varepsilon \rho_f C_{pf} + (1-\varepsilon) \rho_r C_{pr}) \frac{\partial T}{\partial t} = \nabla \cdot (\lambda_r \nabla T) - \rho_f C_{pf} \boldsymbol{U} \cdot \nabla T + \frac{\mu_f}{k} \| \boldsymbol{U} \|^2 - \left( \frac{\partial ln \rho_f}{\partial ln T} \right)_p \frac{Dp}{Dt} \tag{4}$$

where $C_p$ is heat capacity, $\lambda_r$ is the bulk thermal conductivity of porous rock. Subscripts of $r$ and $f$ refer to rock and fluid, respectively. Fluid and rock are assumed to be in local thermal equilibrium ($T = T_r = T_f$) so that the mixture appears on the left-hand side of equation (4). Changes in temperature depend on conductive heat transport, advective heat transport by fluid flow, heat generation by internal friction of the fluid, and pressure-volume work. All fluid properties are functions of both pressure and temperature and are calculated using the IAPWS-IF97 formulation of water and steam properties as implemented

in the freesteam project (Pye, 2010). Further details on the derivation of the governing equations can be found in the appendix of Hasenclever et al. (2014).

## 2.2   Implemented formulation

We solve for pressure (Equation 3), velocity (Equation 1) and temperature (Equation 4) separately. Based on the finite-volume method implemented in OpenFOAM, the primary variables ($p$ and $T$) related equations are discretized on an cell-centroid

computational grid. The transient temperature term in (Equation 3) is evaluated explicitly and is treated numerically as a source term resulting in a Poisson-type equation

$$\varepsilon \rho_f \beta_f \frac{\partial p}{\partial t} - \varepsilon \rho_f \alpha_f \frac{\partial T}{\partial t} + \nabla \cdot \left( \rho_f \frac{k}{\mu_f} \rho_f \boldsymbol{g} \right) = \nabla \cdot \left( \rho_f \frac{k}{\mu_f} \nabla p \right) \tag{5}$$

where the left-hand side terms are pressure transient term and Laplacian term (or diffusion term), respectively. The first term on the right-hand side is evaluated explicitly using a known temperature field, and the second term on the right-hand side is a

divergence term of gravity related flux ($\phi_{\boldsymbol{g}}$), which is defined on each face of the computational grid.

To apply the finite-volume method, the advection term (the second term on the right-hand side) in the temperature equation (Equation 4) should be reformulated as a divergence term

$$\rho_f C_{pf} \boldsymbol{U} \cdot \nabla T = \nabla \cdot (\rho_f C_{pf} \boldsymbol{U} T) - T \nabla \cdot (\rho_f C_{pf} \boldsymbol{U}) \tag{6}$$





Then substituting equation (6) in equation (4), the temperature equation can be rearranged as

$$130 \quad (\varepsilon\rho_f C_{pf} + (1-\varepsilon)\rho_r C_{pr})\frac{\partial T}{\partial t} + \nabla\cdot(\rho_f C_{pf}\boldsymbol{U}T) = \nabla\cdot(\lambda_r\nabla T) + T\nabla\cdot(\rho_f C_{pf}\boldsymbol{U}) + \frac{\mu_f}{k}\parallel\boldsymbol{U}\parallel^2 + T\alpha_f\frac{Dp}{Dt} \qquad (7)$$

where on the left-hand side, the first term is temperature transient term and the second one is the advection term. On the right-hand side, the first two terms represent temperature diffusion and the source term resulting from the re-formulation given in eq. (6), respectively. The last two source terms are calculated explicitly.

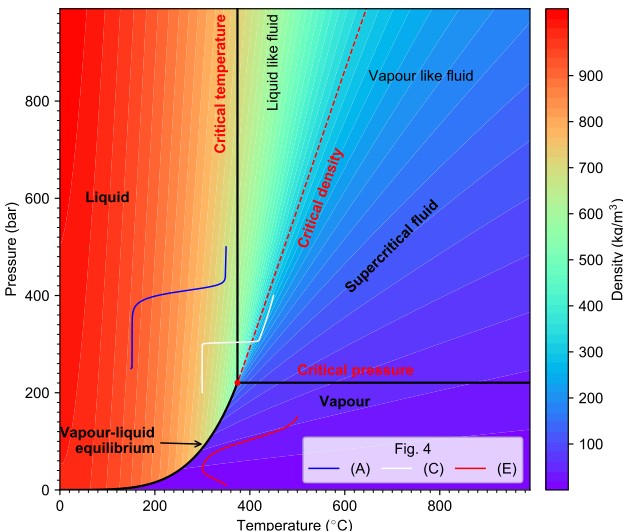

**Figure 1.** Phase diagram and density of pure water in temperature-pressure space. Red dot denotes supercritical point. The supercritical fluid region is outlined by critical pressure and critical temperature lines, and is divided by critical density isoline into liquid-like fluid and vapour-like fluid. Three solid curves in different colors represent pressure-temperature path of 1-D benchmark examples shown in figure 4 (A), (C) and (E), respectively.

### 2.3 Time-step limitations

To determine the time step, we adopt the limitation related to the Courant number Co, which is defined for compressible fluid as

$$Co = \max_{\forall cell}\left(0.5\frac{\sum_{F=0}^{m}\phi_i}{V_{cell}}\right)\Delta t \qquad (8)$$

with $m$ being the number of neighbor faces $F$ to a specific cell. Then the coefficient for time-step change is written as

$$C_{\Delta t} = \frac{Co_{fixed}}{Co} \qquad (9)$$

To avoid too large changes of the time-step which could lead to numerical instabilities, we adopt the time-step as follows (Jasak,
1996)

$$\Delta t = min(min(C_{\Delta t}, 1 + 0.1C_{\Delta t}), 1.2)\Delta t_{last} \qquad (10)$$





Implementation details can be found in the OpenFOAM documentation and the OpenFOAM source files included by the main source code file HydrothermalSinglePhaseDarcyFoam.C.

### 2.4 Boundary conditions

To solve the pressure and temperature equations, we have to impose suitable boundary conditions for $T$ and $p$. The "typical" boundary conditions, e.g. fixed value, fixed gradient and mixing of both, are directly inherited from the basic boundary conditions of OpenFOAM. In submarine hydrothermal system modeling, also some special adaptations of these basic boundary conditions can be useful.

The hydrothermal heat flux ($q_h$) boundary condition is a fixed gradient boundary condition that is often used to approximate

heat input from a crustal magma chamber and is commonly used for simulations of mid-ocean ridge hydrothermal systems (e.g. Coumou et al., 2009; Weis et al., 2014). This Neumann boundary condition is called *hydrothermalHeatFlux* in the toolbox and can be used for the temperature field. Using it, the imposed gradient of temperature can be expressed as

$$\boldsymbol{n} \cdot \nabla T = -\boldsymbol{n} \cdot \frac{\boldsymbol{q_h}}{\lambda_r} \tag{11}$$

where $\boldsymbol{n}$ denotes the normal vector of the face boundary. In addition, we implement two options for heat flux distribution. Using

the keyword shape allows modifying the functional form of the heat flux boundary. The available options are fixed, gaussian2d, gaussian3d. The default option is fixed, and if gaussian2d or gaussian3d is specified, the Gaussian shape (Equation 12) related parameters (qmin, qmax, c, x0 and/or z0) have to be specified (see subsection 5.3).

$$q_h(x,z) = q_{min} + (q_{max} - q_{min})e^{-\frac{(x-x_0)^2+(y-z_0)^2}{2c^2}} \tag{12}$$

A similar boundary condition called *hydrothermalMassFluxPressure* is defined for the pressure field, since it is not straightfor-

160 ward to impose fluid velocities on boundaries. This Neumann boundary condition for pressure can be used to prescribe a mass influx into the modeling domain ($\boldsymbol{\phi_m} = \rho_f \boldsymbol{U}$). The corresponding gradient of the pressure field can be derived from Darcy's law (Equation 1)

$$\boldsymbol{n} \cdot \nabla p = \boldsymbol{n} \cdot \left(\rho_f \boldsymbol{g} - \frac{\mu_f}{k}\boldsymbol{U}\right) = \frac{\mu_f}{\rho_f k}\boldsymbol{n} \cdot (\boldsymbol{\phi_g} - \boldsymbol{\phi_m}) \tag{13}$$

Where $\boldsymbol{\phi_m}$ and $\boldsymbol{\phi_g}$ denote mass flux and gravity related flux, respectively. Further, we define another Neumann boundary

condition (named *noFlux*) of pressure field for impermeable boundaries, which is a special case of HydrothermalMassFlux-Pressure when $\phi_m = 0$. In addition, a Dirichlet boundary condition (named *submarinePressure*) for pressure field is defined to describe hydrostatic pressure at the seafloor boundary due to bathymetric relief. Another commonly used boundary condition on hydrothermal venting boundary (e.g. seafloor) is OpenFOAM's *inletOutlet* boundary conditions, which allows to set a constant temperature for inflow and zero heat flux for outflowing nodes - this type of boundary conditions is often used to mimic free

venting at the seafloor.





## 2.5 Fluid properties and equation-of-state

Numerical solutions of hydrothermal flow are known to strongly depend on the used thermodynamic properties of the simulated fluid. A series of studies using realistic thermodynamic properties of pure and salt water, rather than making a Boussinesq approximation or using linearized properties, have shown that realistic results depend critically on using a realistic EOS (Jupp

and Schultz, 2000; Hasenclever et al., 2014; Driesner, 2010; Carpio and Braack, 2012). Note that we here do not address any issues related to using pure versus saltwater EOSs, as outlined in the introduction. We use an EOS for pure water based on the IAPWS-IF97 parameterization and have created a corresponding OpenFOAM thermo-physical model. The phase diagram is shown in Figure 1.

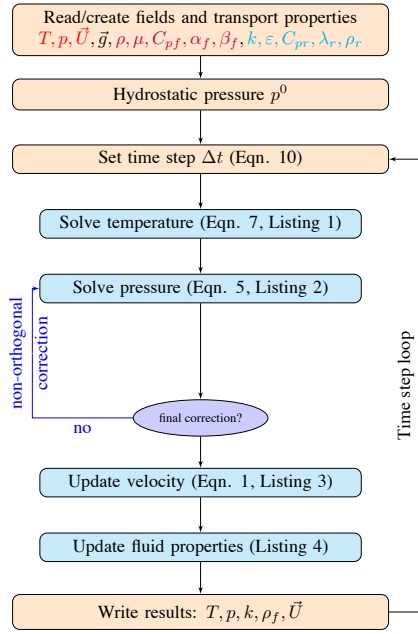

**Figure 2.** Schematic of the sequential algorithm.

## 2.6 Solution algorithm

The governing equations of pressure and temperature are solved in a sequential approach. The primary variables (pressure and temperature) and transport properties (such as permeability, porosity, etc.) have to be initialized before the time loop, and then the initial Darcy velocity and thermodynamic properties of fluid can be updated according to the temperature and pressure fields. The main computational sequence for a single time step are described below and sketched in Figure 2.

1. The time step size $\Delta t^{n+1}$ is calculated from the Courant number related condition (Equation 8).





2. Temperature field $T^{n+1}$ is implicitly computed by solving the energy conservation equation (Equation 7). The syntax of a partial differential equation (PDE) in OpenFOAM is very closed to mathematical formulation and a code snippet of the temperature equation 7 implementation is shown in listing 1. All variable symbols, names and OpenFOAM types (classes) are shown in Table 1. The transient term $\partial T/\partial t$ can be implicitly discretized using the OpenFOAM operator fvm::ddt(T) with the discretization scheme (e.g. Euler scheme) being specified under the keyword ddtSchemes in the system/fvSchemes

dictionary file (see Listing 9). The divergence term, Laplace term and source term are implicitly discretized using fvm::div, fvm::laplacian and fvm::Sp operators. The last term on the right-hand side is explicitly calculated using known field values from the current or previous time step; the corresponding time derivative and gradient can be programmed using fvc::ddt and fvc::grad, respectively.

---

**Listing 1** Implementation of temperature equation 7 with OpenFOAM (in EEqn.H).

```
fvScalarMatrix TEqn(
    (porosity*rho*Cp+(1.0-porosity)*rho_rock*cp_rock)*fvm::ddt(T)
    +fvm::div(phi*fvc::interpolate(Cp),T)
    ==
    fvm::laplacian(kr,T) + fvm::Sp(fvc::div(phi*fvc::interpolate(Cp)),T)
    + mu/permeability*magSqr(U) + T*alphaP*(fvc::ddt(p)+(U & fvc::grad(p)))
);
TEqn.solve();
```

---

   3. The pressure field $p^{n+1}$ is implicitly computed by solving the pressure equation (Equation 5), the code snippet is shown

in Listing 2. The temperature temporal term and divergence of $\phi_g$ on the right-hand side are evaluated explicitly by using fvc::ddt(T) and fvc::div(phig) (see line 7 in Listing 2). Although pressure boundary conditions are customized by flux directly (see subsection 2.4), in order to specify pressure boundary conditions through velocity boundary conditions, e.g. OpenFOAM's fixedFluxPressure boundary condition, the OpenFOAM's function of constrainPressure has to be called before solving pressure equation (see line 3 in Listing 2). For non-orthogonal mesh, a non-orthogonal correction algorithm

(line 4 in Listing 2) is commonly adopted to improve accuracy for gradient computation. The number of non-orthogonal correction is specified by nNonOrthogonalCorrectors key in PIMPLE sub-dictionary in system/fvSolution file.





---

**Listing 2** Implementation of pressure equation 5 with OpenFOAM (in pEqn.H).

```
surfaceScalarField rhorAUf("rhorAUf", fvc::interpolate(rho*permeability/mu));
surfaceScalarField phig("phig",(fvc::interpolate(rho)*rhorAUf * g) & mesh.Sf());
constrainPressure(p, rho, U, phig, rhorAUf);
while (pimple.correctNonOrthogonal()){
    fvScalarMatrix pEqn(
        porosity*rho*betaT*fvm::ddt(p) - fvm::laplacian(rhorAUf,p)
        -porosity*rho*alphaP*fvc::ddt(T) + fvc::div(phig)
    );
    pEqn.solve();
}
```

---

4. The velocity field is calculated explicitly using latest pressure field based on Darcy's law (Equation 1). Instead of calculating the velocity directly, we implement an indirect approach based on OpenFOAM's function fvc::reconstruct to reconstruct the velocity field from the computed mass flux (see Listing 3), which performs higher numerical stability

and benefits from the flux conservation characteristics of the finite volume method. In addition, boundary conditions of velocity field have to be updated (line 3 in Listing 3) if OpenFOAM's fixedFluxPressure boundary condition is applied for pressure field.

---

**Listing 3** Implementation of Darcy velocity calculation with OpenFOAM (in pEqn.H).

```
phi = phig + pEqn.flux();
U = permeability/mu*fvc::reconstruct(phi/rhorAUf);
U.correctBoundaryConditions();
```

---

5. Thermodynamic properties of fluid are updated by the thermo-physical model after solving temperature and pressure field. The implementation code snippet is shown in Listing 4, in which thermo.correct() is used to update temperature and

210 pressure value for all the calculating nodes. Then the thermodynamic propertie of fluid, for example density ($\rho$), at each nodes are calculated based on IAPWS-IF97 (see line 2-6 in Listing 4).





---

**Listing 4** Update fluid thermo dynamic properties (in updateProps.H).

thermo.correct();

rho=thermo.rho();

mu=thermo.mu();

Cp=thermo.Cp();

alphaP=thermo.alphaP();

betaT=thermo.betaT();

---

## 2.7 Numerical schemes

Since the numerical evaluation of the divergence and gradient terms in the governing equations has great influence on heat and mass transfer, a suitable solution strategy regarding discretization and linear solver schemes need to be chosen to ensure
accuracy, robustness and stability. In the presented solver *HydrothermalFoam*, the discretization and interpolation scheme of the primary fields $(T, p)$ can be defined in the simulation configuration files. In the following benchmark tests section 5, the advective discretization scheme is upwind to ensure consistency with results of other software packages. It should be noted that the generic implementation of *HydrothermalFoam* solver allows all of the commonly used numerical schemes of OpenFOAM (Jasak, 1996) and using a high-order advection scheme like e.g. Van Leer is often the better choice.

## 220 3 Description of toolbox components

The organization of the *HydrothermalFoam* toolbox is shown in Figure 3. The toolbox contains 5 parts: *HydrothermalFoam* solver, thermophysical models, boundary conditions, cookbooks and manual.

- HydrothermalSinglePhaseDarcyFoam: this block compiles the solver (an executable file) that solves the seafloor hydrothermal convection equations described in subsection 2.1. It can be used to simulate single-phase hydrothermal circulation in
an isotropic porous medium.

- ThermoModels: this block compiles the libPureWaterThermophysicalModels library containing the EOS of pure water, which is used to formulate the used thermophysical model - see subsection 2.5.

- BoundaryConditions: this block compiles libHydrothermalBoundaryConditions library containing four customized boundary conditions explained in subsection 2.4. The example usage of each boundary conditions can be found in
cookbooks and manual in GitLab repository (https://gitlab.com/gmdpapers/hydrothermalfoam).

- benchmarks: input files of all the benchmark tests (see section 5) presented in this paper.

- cookbooks: this block contains some example cases of parallel computing, user defined boundary conditions, and post processing.





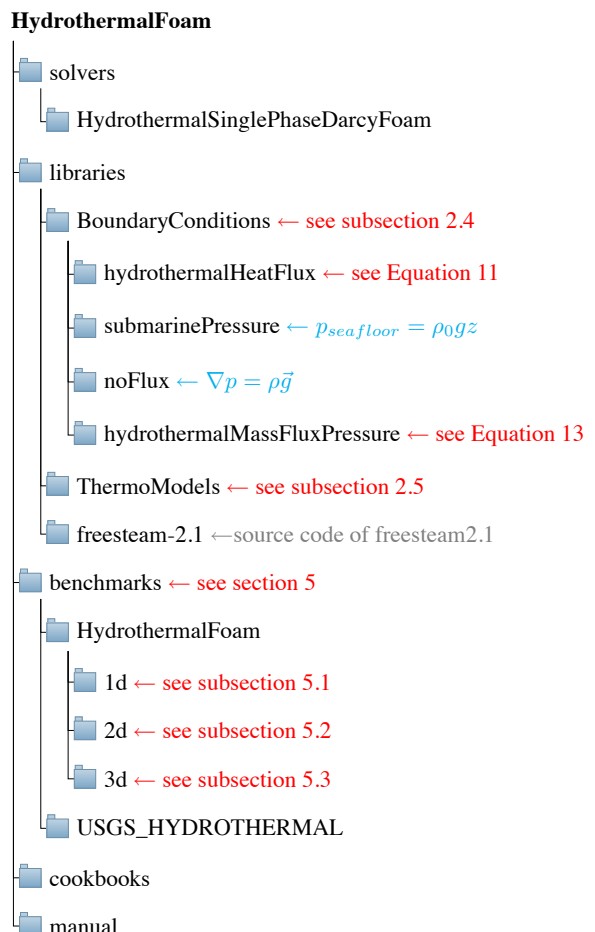

**Figure 3.** Structure and components of the *HydrothermalFoam* toolbox.

– manual: the manual present detailed usage and all the numerical examples in cookbooks.

**4  Installation**

We provided two options for installation: one is building from source and the other is using a precompiled docker image.





### 4.1 Building from source code

#### 4.1.1 OpenFOAM

The *HydrothermalFoam* v1.0 is developed based on OpenFOAM-7, which can be installed according to the installation
instructions (https://openfoam.org/download/) given by the development team for Ubuntu Linux, Other Linux, macOS and
Windows platform, respectively.

#### 4.1.2 *HydrothermalFoam*

Once OpenFOAM is built successfully, the source code of *HydrothermalFoam* can be downloaded from Zenodo.org (Guo and
Rüpke, 2020). The directory structure and components of *HydrothermalFoam* are shown in Figure 3 and the components can be
built follow three steps below,

1. Build freesteam-2.1 library. The freesteam project is constructed by scons, which is a open source software construction
   tool dependent on python 2, and based on GSL (**G**NU **S**cientific **L**ibrary). Therefore python 2, scons and GSL have to be
   installed firstly, then change directory to freesteam-2.1 in *HydrothermalFoam* source code and type command of scons
   install to compile freesteam library.

2. Build libraries of customized boundary conditions and thermo-physical model. Change directory to libraries and type
   command of ./Allmake to compile the libraries.

3. Build solver of *HydrothermalSinglePhaseDarcyFoam*. Change directory to HydrothermalSinglePhaseDarcyFoam and
   type command of wmake to compile the solver.

All the library files and executable application (solver) file will be generated in directories defined by OpenFOAM's path
variables of FOAM_USER_LIBBIN and FOAM_USER_APPBIN, respectively.

### 4.2 Precompiled docker image

In order to use all the tools directly without any compiling and development skills, we have published a precompiled Docker®
image in DockerHub repository of zguo/hydrothermalfoam. The docker image can be used on any operation systems (e.g.
Windows, Mac OS and Linux) to run *HydrothermalFoam* cases follow five steps below,

1. Install Docker, then open Docker and keep it running.

2. Pull the docker image by using command of docker pull zguo/hydrothermalfoam in shell terminal, e.g. bash shell in Mac
   OS, PowerShell in Windows system.

3. Install a container from the docker image by running shell script, e.g. Unix shell script, shown in Listing 5. The directory
   named HydrothermalFoam_runs is a shared folder between the container and host machine.



---

**Listing 5** Script for installing a container from the *HydrothermalFoam* docker image.

---

dirInContainer="/home/openfoam/HydrothermalFoam_runs"

homeInHost="${1:-$HOME}"

dirInHost="${homeInHost}/HydrothermalFoam_runs"

imageName="zguo/hydrothermalfoam"

containerName="hydrothermalfoam"

docker run -it -d --name ${containerName} --workdir="/home/openfoam" -v=${dirInHost}:${dirInContainer}
↪    ${imageName}

---

265    4. Start the container by running command of docker start hydrothermalfoam.

5. Attach the container by running command of docker attach hydrothermalfoam. The user now in a Ubuntu linux environment
with precompiled *HydrothermalFoam* tools which located at directory of  /HydrothermalFoam. We recommend user
run *HydrothermalFoam* cases in the directory of HydrothermalFoam_runs in the container, and then the results are
synchronized in the shared directory in the host, and thus can be visualized by ParaView®, Tecplot® or other software.

270 Following above five steps, one can run *HydrothermalFoam* tools in the current shell terminal. detailed instructions and
five-minutes tutorial video can be found in the DockerHub repository.

### 4.3    Run the first case of *HydrothermalFoam*

The basic directory structure for a *HydrothermalFoam* case, that contains the mandatory files to run an application, is shown in
Figure 4. There is a bash script file named run.sh in every *HydrothermalFoam* cases provided by this paper. In addition, we
275 provide a five-minutes quick start tutorial video to run the first case of *HydrothermalFoam* in Docker. A *HydrothermalFoam*
case can be run by executing ./run.sh.

#### 4.3.1    Mesh generation

The mesh information containing boundary patches definitions, cell face indices and connections is located in polyMesh
subdirectory in constant directory in a specific case folder (Figure 4). All the OpenFOAM mesh generation approaches can be
280 applied to *HydrothermalFoam* as well. For example, blockMesh generates a simple mesh defined by blockMeshDict dictionary
file in system directory, and gmshToFoam transform a mesh file generated by Gmsh (Geuzaine and Remacle, 2009) to polyMesh.

#### 4.3.2    Input fields data

Much of the input/output data in *HydrothermalFoam* are fields, e.g. temperature, pressure data, that are read from and written
into the time directories. For example, the initial time directory is commonly named 0 (see Figure 4). HydorthermalFoam writes
285 field data as dictionary files using keyword entries described in Table 2. The required input field data of *HydrothermalFoam* are



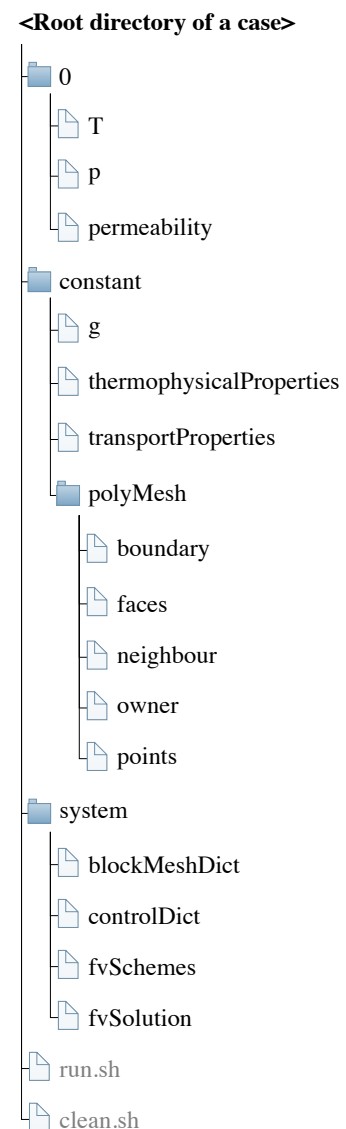

**Figure 4.** Case directory structure of the *HydrothermalFoam* toolbox.

temperature, pressure and permeability. Each input field data begins with an entry for its dimensions, which is expressed by a vector of seven basic SI units (International System of Units) in order of $[kg, m, s, K, mol, A, cd]$. The file names are the same as variable names shown in Table 1. Following that, is the internalField, described in one of the following ways.

1. **Uniform field**. A single value is assigned to all elements within the field, taking the form:

internalField uniform <entry>;





**Table 2.** Main keywords used in field dictionary files

| Keyword | Description | Example |
|---------|-------------|---------|
| dimensions | Dimension of field | [0 0 0 1 0 0 0] |
| internalField | Value of internal field | uniform 278.15 |
| boundaryField | Boundary field | see Listing 6 |

2. **Nonuniform field**. Each field element is assigned a unique value from a list, taking the form:

   internalField nonuniform <List>;

   The nonuniform list of internal field, e.g. permeability, can be specified in different mesh regions by using setFields transform setFieldsDict in system directory (see the Heterogeneous benchmark example described in section 5.3.2).

---

**Listing 6** Example dictionary file for temperature field T

---

```
dimensions      [0 0 0 1 0 0 0];
internalField   uniform 278.15;
boundaryField
{
    sides { type zeroGradient;}
    top { type inletOutlet; phi phi; inletValue uniform 278.15;}
    bottom { type fixedValue; value uniform 873.15;}
    frontAndBack { type empty;}
}
```

---

The boundaryField is a dictionary containing a set of entries whose names are listed in the polyMesh/boundary file for each boundary patches. The compulsory entry, type, describes the boundary condition specified for the field. Besides OpenFOAM internal basic boundary condition type of fixedValue, zeroGradient, codedFixedValue, et al., the customized boundary condition types, e.g. noFlux for pressure and HydrothermalHeatFlux for temperature, are described in subsection 2.4. Because permeability field is not solved but the boundaryField key has to be set, therefore type of all the boundary patches for permeability is set to

zeroGradient for any cases. An example set of field dictionary entries for temperature T are shown in Listing 6.





### 4.3.3 Thermophysical models and transport properties

The transportProperties and thermophysicalProperties are two compulsory files in constant subdirectory. The first one contains constant properties of rock, e.g. porosity, density, are shown in Listing 7, and the second one contains keywords (Listing 8) of the new defined thermophysical model for water which is described in section 2.5.

**Listing 7** transportProperties file

```
porosity porosity [0 0 0 0 0 0 0] 0.1;
kr kr [1 1 -3 -1 0 0 0] 2;
cp_rock cp_rock [0 2 -2 -1 0 0 0] 880;
rho_rock rho_rock [1 -3 0 0 0 0 0] 2700;
```

**Listing 8** thermophysicalProperties file

```
thermoType
{
    type            htHydroThermo;
    mixture         pureMixture;
    transport       IAPWS;
    thermo          IAPWS;
    equationOfState IAPWS;
    specie          specie;
    energy          temperature;
}
mixture
{
    specie
    {
        molWeight       18;
    }
}
```





### 4.3.4 Discretized schemes and solution control

The discretization schemes for primary variables in the PDEs (Partial Difference Equations) and solver for linear equations
are specified in fvSchemes and fvSolution files in system directory, respectively. According to implementation of temperature
equation (Listing 1) and pressure equation (Listing 2), we have to specify discretization schemes for the transient terms, Laplace
terms, and gradient and divergence terms, which are shown in Listing 9. A example of solver, preconditioner and tolerance
settings for linear equations of temperature and pressure fields are shown in Listing 10. We recommend to keep these two files
the same for different cases unless one attempts to try different options available in OpenFOAM.

## 5 Benchmark tests

We have conducted a number of one-dimensional (1-D), two-dimensional (2-D), and three-dimensional benchmark tests and com-
pared the results to other established software packages to validate *HydrothermalFoam* and to highlight some of its advantages.
The reference software we used is version 3.1 of HYDROTHERM, a simulation tool developed and maintained by the US Geo-
logical Survey (USGS), which can be downloaded from the internet (https://volcanoes.usgs.gov/software/hydrotherm/index.html
) for free. All the parameters used in the 1-D and 2-D examples are taken from Weis et al. (2014), who presented a sequence of
well-defined and highly useful benchmarks designed to test code performance within different key thermodynamic fluid states.
In those benchmarks the transport properties and rock properties are constant and uniform (values are also listed in table 1); an
isotropic permeability of $k = 10^{-15} \ m^2$, a porosity of $\varepsilon = 0.1$, a heat capacity of $C_{pr} = 880 J/(kg \ ^\circ\mathrm{C})$, a thermal conductivity
of $\lambda_r = 2 \ W/(m \ ^\circ\mathrm{C})$ and a rock density of $\rho_r = 2700 \ kg/m^3$ are used in all simulations below.

### 5.1 One-dimensional simulations

We conducted six 1-D simulations to test the code performance along the three $p - T$ paths in the phase diagram of pure
water shown in Figure 1. These runs are designed with constant pressure and temperature conditions on both ends of a
domain with 2 km length and 10 m grid spacing. The boundary conditions and initial conditions of each 1-D test are listed
in table 2. For comparison, we use the same parameters for the HYDROTHERM simulations. The computational domain
is oriented horizontally (model index are A, C, E) without gravity and vertically (model index are B, D, F) with gravity
to evaluate gravitational effects on fluid flow. All input files can be be found in the bechmarks/HydrothermalFoam/1d and
bechmarks/USGS_HYDROTHERMAL/1d directories.

Simulation results of the six 1-D examples are shown in figure 5. The example runs A and B describe invasion of a hot fluid
into an initially colder domain and the fluids stay in single-phase liquid state. The thermal front moves from the start point
(left or bottom for the horizontal and vertical example, respectively) towards the end point. As the vertical flow is opposing
gravity, the flow is about three times slower. The fluids remain at pressures beyond the critical end point of pure water and
therefore in the single-phase regime. Results calculated by HYDROTHERM and *HydrothermalFoam* are almost identical. In
Examples A (horizontal) and B (vertical), the fluid remains in a liquid-like state, while in the examples C (horizontal) and





**Table 3.** Model parameters of 1-D benchmark examples

| Model index | Boundary conditions | | Initial conditions | End time (year) | Orientation |
|:---:|:---:|:---:|:---:|:---:|:---:|
| | left/bottom | right/top | | | |
| A | 300 °C , 50 MPa | 150 °C , 25 MPa | 150°C , 25 MPa | 250 | Horizontal |
| B | 300 °C , 50 MPa | 150 °C , 25 MPa | 150°C , 25 MPa | 250 | Vertical |
| C | 450 °C , 40 MPa | 300 °C , 20 MPa | 300°C , 20 MPa | 250 | Horizontal |
| D | 450 °C , 40 MPa | 300 °C , 20 MPa | 300°C , 20 MPa | 250 | Vertical |
| E | 500 °C , 15 MPa | 350 °C , 1 MPa | 350°C , 1 MPa | 250 | Horizontal |
| F | 500 °C , 15 MPa | 350 °C , 1 MPa | 350°C , 1 MPa | 250 | Vertical |

D (vertical) the fluid is flowing along the pressure gradient from a cold liquid-like state to a hot vapor-like state. In C,D the fluids moves about two times faster than in examples A,B resulting in a sharper thermal front. The sharpness of this front is, unfortunately, often affected by the numerical scheme (e.g. mesh geometry, upwinding scheme, and advection scheme). In fact, OpenFOAM seems to cope a bit better with resolving the sharpness of the front despite also using an upwind advection scheme.

Benchmarks E,F explore sub-critical vapor flow. The results of horizontal and vertical flow look very similar because density of single-phase vapor fluid is very low and thus gravitational effect are relatively small with respect to the liquid cases. The results of *HydrothermalFoam* have a good agreement with that of HYDROTHERM for single-phase vapor flow.

## 5.2 Two-dimensional simulations

The two-dimensional models are performed on a rectangular domain with a length of 9 km in the x-direction and 3 km in the y-direction (Figure 6), loosely representative of a vertical section through the upper oceanic crust with uniform permeability. The top boundary represents the seafloor and is kept at a constant pressure of 30 MPa, which is equivalent to about 3 km water depth and a constant temperature of 5°C. At the bottom, a constant heat flux of $Q_b = 0.05 \ W/m^2$ is applied. Further, we assume a magmatic heat source with constant heat flux of $Q_m = 5 \ W/m^2$ extending 1 km wide along x-direction located at the center of the bottom boundary (shown in red line in Figure 6).

The simulation results are shown in Figure 7. Fluid pressure and temperature calculated by *HydrothermalFoam* and HYDROTHERM agree very well. The evolution of the hydrothermal plume is summarized in figure (Figure 7A-C). A hot plume is forming at the base and rises upwards after 5 kyrs (Figure 7A) reaching the seafloor at 15 kyrs (Figure 7B). After 50 kyrs, a quasi-steady hydrothermal circulation cell is established (Figure 7C). The results of *HydrothermalFoam* and HYDROTHERM are again very similar. All input files can be be found in the bechmarks/HydrothermalFoam/2d and bechmarks/USGS_HYDROTHERMAL/2d directories.

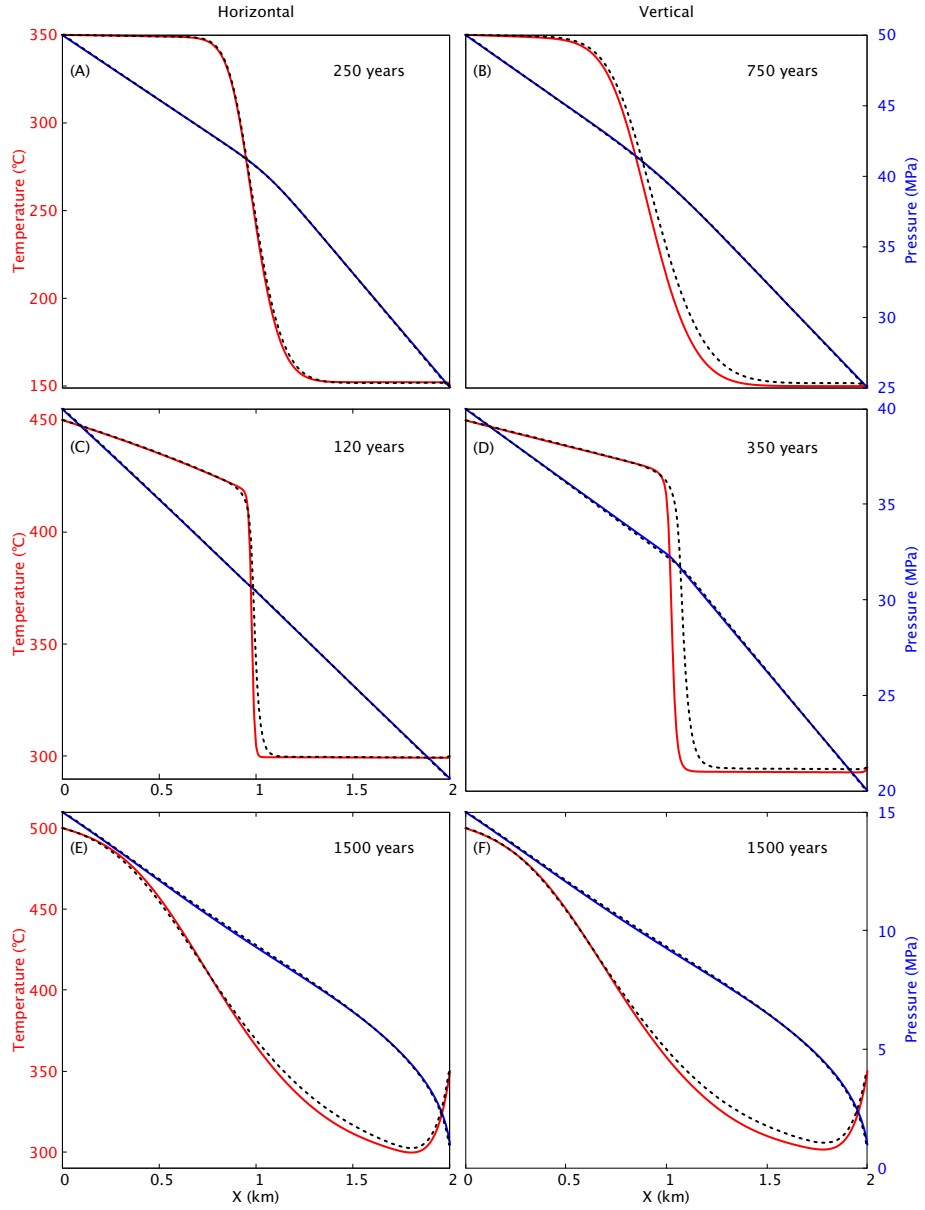

**Figure 5.** Snapshots of one-dimensional benchmark examples in horizontal (A, C, E) and vertical (B, D, F) orientation. Results (temperature in red and pressure in blue) of *HydrothermalFoam* and HYDROTHERM are plotted as solid lines and dashed lines, respectively.

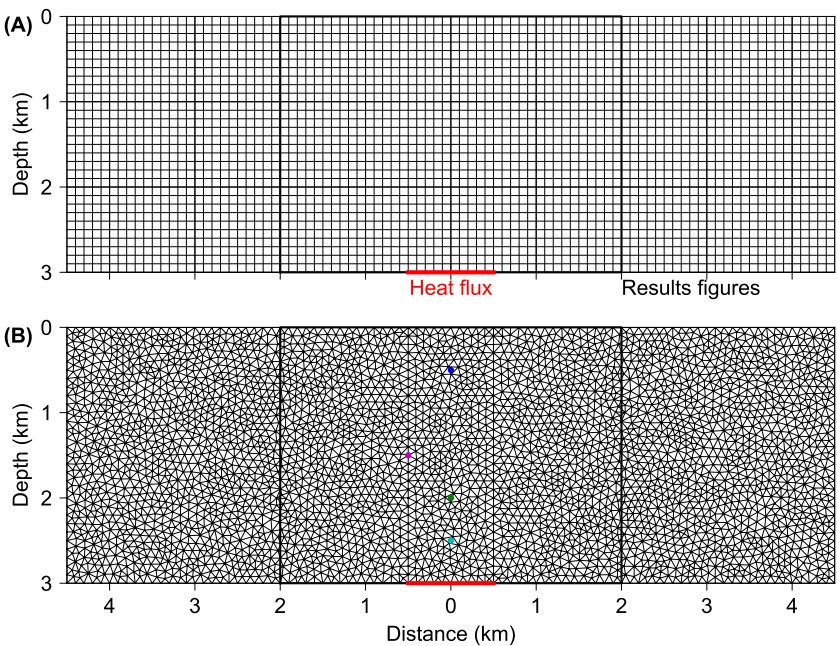

**Figure 6.** Mesh and model configuration of the two-dimensional examples. (A) The grid for HYDROTHERMAL has to be set to a uniform regular mesh with resolution of 100 m and with total number of 2700 elements. (B) *HydrothermalFoam* uses an unstructured mesh with 7074 triangular elements and the edge length in range of $60 \sim 142$ m. Four dots with different color in (B) represent temperature probes location and the black box shows result visualization domain. Red lines on the bottom boundaries denotes heat flux boundary condition for temperature.

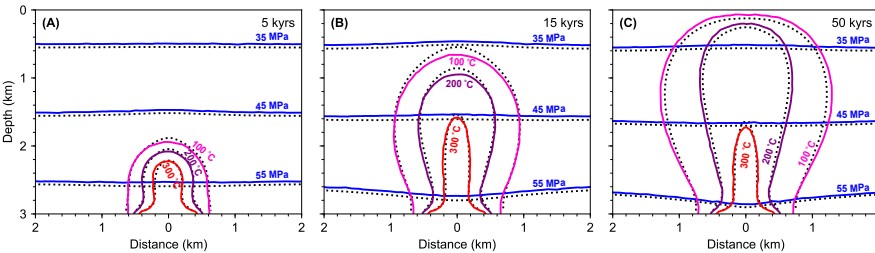

**Figure 7.** Snapshots of two-dimensional examples. Contour lines for fluid pressure (blue) and temperature (red) calculated from *HydrothermalFoam* are plotted as solid lines, and that from HYDROTHER are plotted as dotted lines (black). The visualization window are shown by black box in Figure 6

## 5.3 Three-dimensional simulation

### 5.3.1 Homogeneous model

Similar to the two-dimensional model in subsection 5.2, the three-dimensional model are performed on a cubic domain with a length of 9 km in both the x-direction and the z-direction and 3 km in y-direction (vertical direction). Note that the vertical

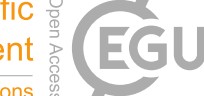

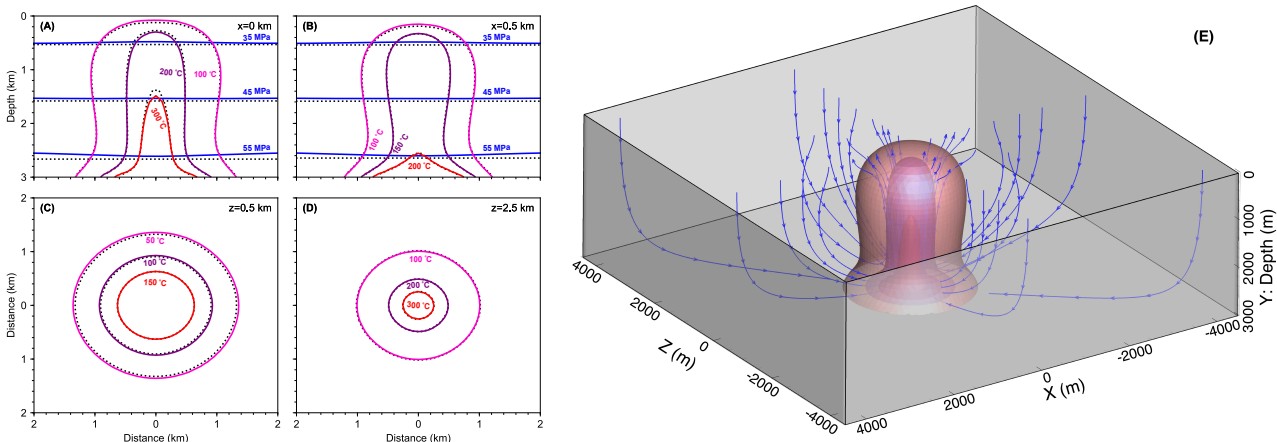

**Figure 8.** Snapshots of three-dimensional model results. Contour lines for fluid pressure (blue) and temperature calculated from *Hydrothermal-Foam* are plotted as solid lines, and that from HYDROTHERM are plotted as black dotted lines. (A) and (B) show pressure and temperature contours on vertical slices, (C) and (D) show temperature contours on horizontal slices. (E) show three-dimensional isothermal surfaces and stream lines.

coordinate in *HydrothermalFoam* or OpenFOAM is $y$ rather than $z$. It can be imagined as representing a three-dimensional section of oceanic crust with uniform permeability. The top boundary is the seafloor and is kept at a constant pressure of 30 MPa and a constant temperature of 5 °C. At the bottom boundary, a zero mass flux is applied for pressure, and a constant Gaussian shaped heat flux (see Equation 12) is applied for temperature. The corresponding parameters in Equation 12 are $x_0 = 0, z_0 = 0, q_{max} = 5\ W/m^2, q_{min} = 0.05\ W/m^2, c = 500$. All input files can be be found in the 3d/Homogeneous directory in bechmarks/HydrothermalFoam and bechmarks/USGS_HYDROTHERMAL directories, respectively.

The simulation results at 50 kyrs are shown in Figure 8. Vertical slices at x=0 km and x=0.5 km, and horizontal slices at z=0.5 km and z=2.5 km are shown in Figure 8(A-D). Fluid pressure (blue contours) and the temperature field calculated by *HydrothermalFoam* and HYDROTHERM (dashed contours) agree very well and results on central vertical slice are very close to two-dimensional model results shown in Figure 7C. Three-dimensional flow path and isothermal surfaces of 300 °C, 200 °C and 100 °C are shown in Figure 8(E).

### 5.3.2 Heterogeneous model

The heterogeneous model with two-layer permeability structure is modified from the homogeneous model described in subsubsection 5.3.1. Permeability of two layers are $k_1 = 10^{-14}\ m^2$ and $k_2 = 10^{-15}\ m^2$, respectively (see Figure 9 A). The thickness of first layer is 1.1 km and the other parameters are the same as the homogeneous model. All input files can be be found in the 3d/Heterogeneous directory in bechmarks/HydrothermalFoam and bechmarks/USGS_HYDROTHERMAL directories, respectively. The simulation results at 50 kyrs are shown in Figure 9. Vertical slices at x=0 km and x=0.5 km, and horizontal





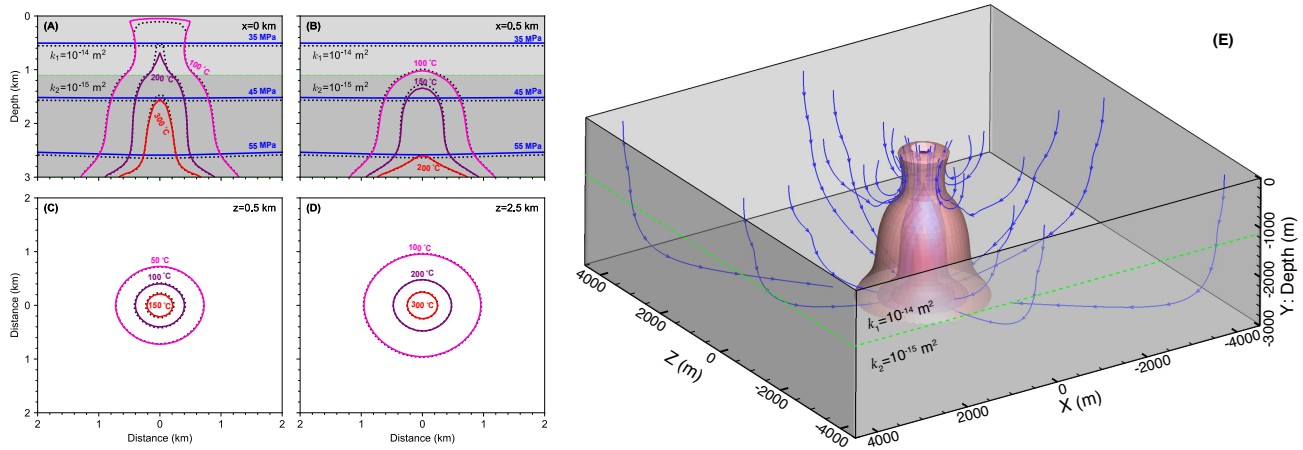

**Figure 9.** Results of three-dimensional model with two-layer permeability structure. Two layers are separated by the green dashed line. Contour lines for fluid pressure (blue) and temperature calculated from *HydrothermalFoam* are plotted as solid lines, and that from HYDROTHERM are plotted as black dotted lines. (A) and (B) show pressure and temperature contours on vertical slices, (C) and (D) show temperature contours on horizontal slices. (E) show three-dimensional isothermal surfaces and stream lines.

slices at z=0.5 km and z=2.5 km are shown in Figure 9(A-D). The higher permeability in the upper layer results in mixing with colder ambient fluids and focussing of the upflow zone. Three-dimensional isothermal surfaces of 300 °C, 200 °C and 100 °C are shown in Figure 9(E).

### 5.4 Cookbooks

In addition to the presented benchmarks, we have added a number of cookbooks to the code repositories that can be used as starting points for more complex models. These include simple 2-D and 3-D box models, 2-D single-pass loop models, and time-dependent permeability models. They also include examples of how to use more complex meshed generated with e.g. the gmsh mesh generator. We intend to add additional cookbooks in the future and hope to receive contributions from users of *HydrothermalFoam*.

### 6 Conclusions

We have presented a toolbox for simulating flow in submarine hydrothermal circulation systems. Being based on the widely used fluid-dynamic simulation platform OpenFOAM, the toolbox provides the user with robust parallelized 3-D solvers and a whole suite of pre- and post-processing tools. The toolbox is meant to provide the interdisciplinary submarine hydrothermal systems community with an accessible and easy-to-use open-source platform for testing ideas on how hydrothermal systems 'work'. The benchmark tests have shown that model matches previously published models and the cookbooks provide the user with



starting points for building more sophisticated models. By following an open-source approach and by providing extensive code documentation, we hope that the presented model will facilitate integrative studies that combines models with data to better assess the role of submarine hydrothermalism in the Earth System.

*Code availability.*

- – Program title: HydrothermalFoam

- – Source code and docker image on Zeondo.org: https://doi.org/10.5281/zenodo.3755647

- – Source code repository on GitLab: https://gitlab.com/gmdpapers/hydrothermalfoam

- – Precompiled Docker image on Docker Hub: zguo/hydrothermalfoam

– Five-minutes quick start tutorial video: https://youtu.be/6czcxC90gp0

- – Source code documentation: https://www.hydrothermalfoam.info/doxygen

- – Online manual: https://www.hydrothermalfoam.info/manual

- – Licensing provision: GNU General Public License 3.0

- – Programming language: C++

– Nature of problem: Seafloor hydrothermal circulation

- – Solution method: The numerical approach is based on the finite-volume method (FVM).

**Appendix A:  Key code snippets**

*Author contributions.*  ZG, LR and CT designed the project. ZG developed the source code, ran simulations and wrote this paper. LR provided suggestions for the benchmarks and co-wrote the paper and manual. CT provided further suggestions for the manuscript and the model. All
authors discussed and contributed to the final paper.

*Competing interests.*  The authors declare that they have no conflict of interest.





---

**Listing 9** Example of fvSchemes file

---

```
ddtSchemes
{
    default        Euler;
}
gradSchemes
{
    grad(p)        Gauss linear;
    grad(T)        Gauss linear;
}
divSchemes
{
    div(phi,T)     Gauss upwind;
    div((phi*interpolate(Cp)),T) Gauss upwind;
}
laplacianSchemes
{
    laplacian(kr,T) Gauss linear corrected;
    laplacian(rhorAUf,p) Gauss linear corrected;
}
interpolationSchemes
{
    default        linear;
}
snGradSchemes
{
    default        corrected;
}
fluxRequired
{
    default no;
    p;
}
```

---



---

**Listing 10** Example of fvSolution file

---

```
solvers
{
    p
    {
        solver          PCG;
        preconditioner  DIC;
        tolerance       1e-12;
        relTol          0;
    }
    pFinal
    {
        $p;
        relTol          0;
    }
    T
    {
        solver          PBiCG;
        preconditioner  DILU;
        tolerance       1e-06;
        relTol          0;
    }
    "(T)Final"
    {
        $T;
        relTol          0;
    }
}
PIMPLE
{
    momentumPredictor no;
    nOuterCorrectors 1;
    nCorrectors     2;
    nNonOrthogonalCorrectors 0;
}
```

---





*Acknowledgements.* This work was supported by National Key R&D Program of China under contract NO. 2018YFC0309901, 2017YFC0306603, 2017YFC0306803 and 2017YFC0306203, COMRA Major Project under contract No. DY135-S1-01-01 and No. DY135-S1-01-06. The work is also part the IODP-SPP 527 funded by the German Science Foundation (DFG) under project number 428603082.



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
