# Peer review of "HydrothermalFoam* v1.0: a 3-D hydro-thermo-transport model for natural submarine hydrothermal systems"

_Geoscientific Model Development, 2020_

## Author Comment (AC1) · 20 Jul 2020

**1   Technical corrections**

We have discovered a number of technical issues in the original version of HydrothermalFoam v1.0.0. For the convenience of the reviewers and early users of *HydrothermalFoam*, we have fixed those issues and document the changes in this "short comment". In particular, we have fixed a few issues in the benchmark and cookbook cases, both in the docker container and in the gitlab source directory, and have have updated the *thermoPhysical model*. The details are described below.

**1.1 Technical issues of HydrothermalFoam v1.0.0**

We have discovered a few technical issues in the first version of *HydrothermalFoam* which is linked in the discussion paper.

- Mesh generation using gmsh

  The original run.sh of the 2D benchmark example was missing a command to generate the mesh using gmsh. As a consequence, the run.sh script returned an error. This issue was fixed by adding the respective gmsh command (gmsh gmsh/box.geo -3 -o gmsh/box.msh -format msh22) to run.sh, see the new version of run.sh.

- Bash syntax compatibility issues

  Some of the run.sh scripts in the cookbook and benchmark directories contain a command (.    clean.sh) to clean the tutorial case, e.g. cookbooks/2d/Regular2DBox. This command works under our Mac OS test environment but not in the docker container or other linux systems. Therefore, this command was changed to ./clean.sh.

- 0/permeability file missing

  The permeability field input file is missing in the **0** directory in some cases, e.g. benchmarks/HydrothermalFoam/2d. This is due to 0/permeability being incorrectly written into the .gitignore file in the case directory. This has been fixed in the latest version of HydrothermalFoam.

**1.2 Updates to the thermoPhysical model**

The thermoPhysical model implements the pure water equation-of-state and is also the natural place where all other material properties, i.e. the matrix properties of the

porous media, should be implemented. However, the original version v1.0.0 used the constant/transportProperties file to implemented the matrix properties (*porosity, kr, cp_rock, rho_rock*). In this update, we have implemented all porous media properties into the thermoPhysical model. Therefore, in v1.0.1 or the latest version, the transportantProperties file is no longer needed by the *HydrothermalSinglePhaseDarcy* solver. The porous medium properties are inputed as a sub-dictionary (named porousMedia) in constant/thermophysicalProperties. The complete thermophysicalProperties file of the latest version is listed below,

```
FoamFile
{
    version     2.0;
    format      ascii;
    class       dictionary;
    location    "constant";
    object      thermophysicalProperties;
}
thermoType
{
    type            htHydroThermo;
    mixture         pureMixture;
    transport       IAPWS;
    thermo          IAPWS;
    equationOfState IAPWS; //Boussinesq
    specie          specie;
    energy          temperature;
}

mixture
```

```
{
    specie
    {
        molWeight        18;
    }
    porousMedia
    {
        cp_rock cp_rock [0 2 -2 -1 0 0 0] 880;
        rho_rock rho_rock [1 -3 0 0 0 0 0] 2700;
        porosity porosity [0 0 0 0 0 0 0] 0.1;
        kr kr [1 1 -3 -1 0 0 0] 2;
    }
}
```

See also the latest version of manual.

**1.3   Change log of HydrothermalFoam v1.0.1**

1. Fixed few issues in benchmarks and cookbooks.

2. Update thermoPhysical model and deprecated transportantProperties file for *HydrothermalSinglePhaseDarcy* solver.

3. Add serial and parallel computing performance example into cookbooks, see the latest manual for the comparison results.

4. Update docker image with the latest source code.

**2 How to update HydrothermalFoam in the docker container**

The docker image of *HydrothermalFoam* has been updated on **July 19, 2020** and pushed to the Docker Hub repository. Note that it will not be updated frequently in the future, but the *HydrothermalFoam* will be.

There are two ways to update *HydrothermalFoam* source code. The first one is using git to clone and pull the latest version of the *HydrothermalFoam* source code. The second one is using the latest version of the docker image and run the bash script of getHydrothermalFoam_latest.sh in the home directory of the docker container.

We recommend the second approach for users who are not familiar with git. **If you are installing docker image of HydrothermalFoam for the first time**, please just follow the installing instructions in the manual or watch the tutorial video for the beginners to install the docker image and start a docker container. **If you have already installed docker image of HydrothermalFoam before July 19, 2020**, please follow the steps below to use the latest version of HydrothermalFoam.

Open terminal (if Mac OS or Linux) or Windows PowerShell and run commands (text in orange color) as the following steps.

1. Pull the latest version of docker image of *HydrothermalFoam*: docker pull zguo/hydrothermalfoam

2. Check for existing docker containers: docker containers ls –all

3. Remove the **hydrothermalfoam** container if it exist: docker rm hydrothermalfoam

4. Reinstall docker container by running commands in installMacHydrothermalFoam.sh for Mac OS and Linux, or running installWindowsHydrothermalFoam.ps1 for Windows system. See also **installation** section in the manual.

5. Start the docker container of *hydrothermalfoam*: docker start hydrothermalfoam

6. Attach the container of *hydrothermalfoam*: docker attach hydrothermalfoam

7. Getting the latest version of *HydrothermalFoam*: ./getHydrothermal-Foam_latest.sh. Note that the script will download and compile the latest source code automatically. The folder name of the latest source code is **HydrothermalFoam-master** and the previous version existed in the docker container will be renamed as **HydrothermalFoam-master_backup** which can be deleted by user if there is not important user data in the folder.

**3  Bug report**

We keep working on developing and improving the source code and will continue to find bugs. And so will the users of HydrothermalFoam. We would like to encourage all users to submit bug reports using the issue tracker (https://gitlab.com/gmdpapers/hydrothermalfoam/-/issues) of the GitLab repository of HydrothermalFoam.

---

## Editor Comment (EC1) · Thomas Poulet (Editor) · 23 Jul 2020

Dear Zhikui Guo,

Thank you for the notification of this important fixes and update.

Best regards,

Thomas Poulet

---

## Referee Comment (RC1) · Cyprien Soulaine (Referee) · 14 Aug 2020

The paper "HydrothermalFoam v1.0: a 3-D hydro-thermo-transport model for natural submarine hydrothermal systems" describes the development of an open-source code based on the OpenFOAM C++ library for simulating hydrothermal systems using a Darcy model. Overall, the paper is well-organized and well-written. The code benefits from all the OpenFOAM features including parallel computing, three-dimensional domains, polyhedral grids... The validation cases presented in Section 5 are convincing and show the potential of such a toolbox.

The model implemented in HydrothermalFoam is very standard: a compressible single-

phase Darcy flow solver combined with a temperature equation. An important contribution is the integration of IAPWS-IF97 model in OpenFOAM. Although the manuscript does not bring any novelties in terms of modelling (everything is very well-established here), the implementation of the model into a modern and efficient platform can lead to significant progress in the submarine hydrothermal community. In particular, the authors have invested important efforts to make their code accessible to people that are not experts in OpenFOAM and the paper is written as a user manual. In that regard, I think the paper worth publications in Geoscientific Model Development.

I have noted some comments (see below) that should be checked before publication. Some are simply typos, others need to be checked carefully.

Comments:

**l72-74: It is important to mention other initiatives that use OpenFOAM to solve flow and transport in porous media with Darcy-like solvers, e.g. Horgue et al. 2015 and Orgogozo et al. 2015**

Horgue, P. et al., An open-source toolbox for multiphase flow in porous media, Computer Physics Communications, 2015, 187, 217-226

Orgogozo, L. et al., An open-source massively parallel solver for Richards equation: Mechanistic modelling of water fluxes at the watershed scale, Computer Physics Communications, 2014, 185, 3358 - 3371

**l84: I think it is important here to mention that the PDEs are solved implicitly but sequentially.**

**l96: strictly speaking, in fluid mechanics, in laminar flow regime, there is inertia. Darcy's law corresponds to the creeping flow regime when inertia is negligible compared with viscous forces.**

**Eq (2) : the right-hand side (=0) is missing**

**l104: The compressibility of the rock is neglected as soon as the porosity is removed from the time derivative in Eq (1). Then, it is probably better to mention this hypothesis just after Eq (1).**

**the heat conductivity is noted lamba_r in Eqs (4)-(7) but k_r in the code.**

**I am not sure that the energy equation in Eq (4) (and then Eq (7)) is exact. It seems correct only if the continuity equation is a divergence-free equation which is not the case in the paper (this is what appears if you derive the equation from the conservation of the internal energy ddt(e) + div(e U) = div(q) + S with e(p,T) ). Moreover, it is not clear whether the two last terms of the right-hand side are necessary as they seem orders of magnitude lower than the other terms. Can you comment on that? Give an estimation of the weight of these terms and explain in which cases it is important to consider them?**

**Assuming that the equation is correct, the numerical treatment presented in Listing 10 can be improved by making the last term implicit with fvm::Sp( alphaP*(...) , T), which should lead to better numerical stability.**

**l139: "adapt" instead of "adopt"**

**replaced all "can be" by "is/are"**

**l159-160: the sentence "since it is not straightforward to impose fluid velocities on boundaries" is not clear. Actually, here you just transport a velocity value into a boundary condition on the pressure gradient because your solver solves a pressure equation only.**

**l166: what is the difference between submarinePressure and OpenFOAM's Prgh-Pressure?**

**l177: more details will be appreciated on the implementation of the thermo-physical model.**

**l194-1999: the sentence about constrainPressure and fixedFluxPressure is very confusing. Why do you need fixedFluxPressure in your simulations? What is the link with the boundary conditions introduced in Section 2.4?**

**It is difficult for the potential user to know where is the code. It is scattered on too many platforms. The document mentions at least 3 different locations (Zenodo.org, DockerHub, GitLab). In particular, Zenodo.org and GitLab seem to provide the same code.**

**l354-355, l364-365, l374-375: "can be be", "bechmarks"**

**l396 Zenodo.org**

**Section 5: I think it will be interested to have a comparison of the simulation time of HYDROTHERM and HydrothermalFoam. Such a comparison can highlight better the importance of using modern computational platforms.**

---

## Referee Comment (RC2) · Matteo Cerminara (Referee) · 13 Sep 2020

The paper "HydrothermalFoam v1.0: a 3-D hydro-thermo-transport model for natural submarine hydrothermal systems" describes the implementation in the OpenFOAM infrastructure of the Darcy model for a compressible fluid in a porous medium. Fluid and solid media are considered in local equilibrium. The paper describes the corresponding partial differential equations for the balance of mass, momentum, and energy; their boundary conditions; and the equation of state for pure water in a single phase, either in subcritical or supercritical conditions. The algorithm is described in a clear way, taking advantage of the high level C++ programming style used by OpenFOAM. The

code is open-source and multiplatform, thanks to the adoption of Docker. It can easily be downloaded from a number of different repositories. A number of test cases and benchmarks are described clearly and can be used to test the code. I think that the paper is well written and that it is worth of publication, after some necessary revision.

General comments

My main concerns are related to two points: 1) Model formulation. I think that the energy balance equation is not correct; 2) Description of model assumptions and application limits. The assumptions behind the model should be clearly listed. In particular, the physical conditions where the local equilibrium holds should be characterized using the typical time of the dynamics of the process and the thermal equilibrium typical time scale.

Specific comments

**Table 1, definition of the time step: If possible, write times in lowercase, otherwise they can be confused with temperature.**

**Eq. (2): This is not an equation. "= 0" is missing**

**Eq. (4): This equation is correct even in the compressible regime but only if the rock porosity, density and heat capacity are constant in time, and if the fluid heat capacity is constant both in space and time. These assumptions should be stated explicitly in the text. However, I am afraid that the assumptions on the fluid heat capacity are not fulfilled in this paper, because $Cp\_f$ is treated as a VolScalarField, and also because a detailed thermodynamics for water is used (Figure 1, L113 and Listing 4). In this Lagrangian formulation, to correclty consider the variability of the fluid heat capacity, a term proportional to its partial time derivative and another proportional to its gradient should be added.**

**L111: Given a certain resolution, the typical time scale for which this approximation holds should be given. For example, using the data in Table 1, the given rock thermal diffusivity is lambda_r/(Cp_r*rho_r) ∼ 6e-7 m^2/s. This means that, roughly, the typical thermal equilibrium time for a resolution of 100 m is order of 500 years. Thus, for the approximation to hold, the typical time of the dynamics at this resolution should be larger than, roughly, 500 years. In other words, the equilibrium time is related to the natural time step used by the simulation. The regime where this equilibrium assumption holds should be given, to allow the user to use the code in its proper regime of approximations. For an example on this issue, see Remark 2.1 in: Cerminara, M., & Fasano, A. (2012). Modeling the dynamics of a geothermal reservoir fed by gravity driven flow through overstanding saturated rocks. Journal of Volcanology and Geothermal Research, 233–234, 37–54. https://doi.org/10.1016/j.jvolgeores.2012.03.005**

**L123: The description is not following the terms order in Eq. (5). Please correct this.**

**Eq. (6): This reformulation iterates the problem with the spatial variations of the fluid heat capacity, see comment above on Eq. (4) and below on Eq. (7).**

**Eq. (7): The term proportional to the divergence of (rho_f*Cp_f*U) should not be present when the spatial dependence of the fluid heat capacity is correctly taken into account. A term for its temporal dependence is still missing.**

**L171: Here should be clearly stated that the EOS is used in the single phase regime, no phase transition is admitted in this formulation.**

**L119: Usually in OpenFOAM, spatial schemes are first or second order, not high order. Using the upwind interpolation makes the code first order in space. Some word on the time scheme used is also needed.**

---

## Author Comment (AC2) · 21 Sep 2020

As both reviewers had questions about the energy equation, we here provide the full derivation. Most of it is based on the derivation given in Bird (2002) but we here provide some extra information on the individual steps. Starting point is the equation of change of internal energy, which in turn results from subtracting the mechanical energy balance from the full energy balance. It is eqn. 11.2-1 in Bird's book, here written for a porous medium. All the symbols definition and unit are listed in Table 1.

[Figure]

$$\frac{\partial(\varepsilon\rho_f u_f + (1-\varepsilon)\rho_r u_r))}{\partial t} = -\nabla \cdot (\vec{q}) - \nabla \cdot (\rho_f u_f \vec{U}) - p\nabla \cdot \vec{U} + \frac{\mu_f}{k} \parallel \vec{U} \parallel^2 \tag{1}$$

The left-hand side describes the change in internal energy in the porous medium, which is related to conduction (1. term on Rhs), advection (2. term), pressure-volume work (3. term), and viscous dissipation (4. term).

Step 1 is to substitute the thermodynamic identity

Interactive
comment

$$u = h - pV = h - \frac{p}{\rho_f} \tag{2}$$

into eqn. (1):

$$\frac{\partial(\varepsilon\rho_f\left(h_f - \frac{p}{\rho_f}\right) + (1-\varepsilon)\rho_r u_r))}{\partial t} = -\nabla\cdot(\vec{q}) - \nabla\cdot(\rho_f\left(h_f - \frac{p}{\rho_f}\right)\vec{U}) - p\nabla\cdot\vec{U} + \frac{\mu_f}{k}\parallel\vec{U}\parallel^2 \tag{3}$$

Separating the terms and assuming constant porosity yields:

$$\varepsilon\frac{\partial\rho_f h_f}{\partial t} - \varepsilon\frac{\partial p}{\partial t} + (1-\varepsilon)\frac{\partial\rho_r u_r}{\partial t} = -\nabla\cdot(\vec{q}) - \nabla\cdot(\rho_f h_f\vec{U}) + \nabla\cdot(p\vec{U}) - p\nabla\cdot\vec{U} + \frac{\mu_f}{k}\parallel\vec{U}\parallel^2 \tag{4}$$

Little rearrangement:

$$\varepsilon\frac{\partial\rho_f h_f}{\partial t} + (1-\varepsilon)\frac{\partial\rho_r u_r}{\partial t} = -\nabla\cdot(\vec{q}) - \nabla\cdot(\rho_f h_f\vec{U}) + \varepsilon\frac{\partial p}{\partial t} + \vec{U}\cdot\nabla p + \frac{\mu_f}{k}\parallel\vec{U}\parallel^2 \tag{5}$$

Now we open the derivatives for the $\rho_f h_f$ terms:

$$\varepsilon\rho_f\frac{\partial h_f}{\partial t} + \varepsilon h_f\frac{\partial\rho_f}{\partial t} + (1-\varepsilon)\frac{\partial\rho_r u_r}{\partial t} = -\nabla\cdot(\vec{q}) - \rho_f\vec{U}\cdot\nabla h_f - h_f\nabla\cdot(\rho_f\vec{U}) + \varepsilon\frac{\partial p}{\partial t} + \vec{U}\cdot\nabla p + \frac{\mu_f}{k}\parallel\vec{U}\parallel^2 \tag{6}$$

and bring the time derivative of density to the rhs while using mass conservation

$$\varepsilon\frac{\partial\rho_f}{\partial t} + \nabla\cdot(\vec{U}\rho_f) = 0 \tag{7}$$

so that the terms in red drop out, which yields:

$$\varepsilon\rho_f\frac{\partial h_f}{\partial t} + (1-\varepsilon)\frac{\partial\rho_r u_r}{\partial t} = -\nabla\cdot(\vec{q}) - \rho_f\vec{U}\cdot\nabla h_f + \varepsilon\frac{\partial p}{\partial t} + \vec{U}\cdot\nabla p + \frac{\mu_f}{k}\parallel\vec{U}\parallel^2 \tag{8}$$

Equation 8 is the equation of change of internal energy written in term of specific enthalpy - except for the solid term, which we will treat later. The next step is to use the standard thermodynamic identity to move from enthalpy to specific heat (cf. eqn. 9.8-7 in Bird's book):

$$dh_f = \frac{\partial h_f}{\partial T}_p dT + \left(\frac{\partial h_f}{\partial p}\right)_T dp = c_p dT + \left[V - T\left(\frac{\partial V}{\partial T}\right)_p\right] dp \tag{9}$$

Substituting eqn. (9) into (8) results in the specific heat being outside the derivatives (cf. eqn. 11.2-4 in Bird's book):

$$\varepsilon\rho_f c_p\frac{\partial T}{\partial t} + \varepsilon\rho_f\left(\frac{\partial h_f}{\partial p}\right)_T\frac{\partial p}{\partial t} + (1-\varepsilon)\frac{\partial\rho_r u_r}{\partial t} = -\nabla\cdot(\vec{q}) - \rho_f c_p\vec{U}\cdot\nabla T - \rho_f\left(\frac{\partial h_f}{\partial p}\right)_T\vec{U}\cdot\nabla p$$
$$+\varepsilon\frac{\partial p}{\partial t} + \vec{U}\cdot\nabla p + \frac{\mu_f}{k}\parallel\vec{U}\parallel^2 \tag{10}$$

Some re-arrangement using eqn. (9) :

$$\varepsilon\rho_f c_p\frac{\partial T}{\partial t} + (1-\varepsilon)\frac{\partial\rho_r u_r}{\partial t} = -\nabla\cdot(\vec{q}) - \rho_f c_p\vec{U}\cdot\nabla T + \varepsilon\rho_f T\left(\frac{\partial\frac{1}{\rho}}{\partial T}\right)_p\frac{\partial p}{\partial t} + \rho_f T\left(\frac{\partial\frac{1}{\rho}}{\partial T}\right)_p\vec{U}\cdot\nabla p$$
$$+\frac{\mu_f}{k}\parallel\vec{U}\parallel^2 \tag{11}$$

Next we use a logarithmic derivative for the $\frac{\partial \frac{1}{\rho}}{\partial T}$ terms,

$$\rho_f T \left( \frac{\partial \frac{1}{\rho_f}}{\partial T} \right)_p = -\frac{T}{\rho_f} \left( \frac{\partial \rho_f}{\partial T} \right)_p = -\left( \frac{\partial ln\rho_f}{\partial lnT} \right)_p \tag{12}$$

and combine equation 11 (terms in blue and red) with equation 12 to get:

$$\varepsilon\rho_f c_p \frac{\partial T}{\partial t} + (1-\varepsilon)\frac{\partial \rho_r u_r}{\partial t} = -\nabla\cdot(\vec{q}) - \rho_f c_p \vec{U}\cdot\nabla T - \left( \frac{\partial ln\rho_f}{\partial lnT} \right)_p \left( \varepsilon\frac{\partial p}{\partial t} + \vec{U}\cdot\nabla p \right) + \frac{\mu_f}{k} \parallel \vec{U} \parallel^2 \tag{13}$$

which is (almost) the energy equation provided in the main text. The final step is to treat the solid terms. Here we again use the standard thermodynamic identity (cf. eqn. 45.2-45.7 in Feynman et. al., 2011):

$$du = c_v dT + \left[ p - T\frac{\partial p}{\partial T} \right]_p dV \tag{14}$$

As the solid is assumed incompressible, the second term vanishes and $c_v = c_p$, so that

$$(\varepsilon\rho_f c_p + (1-\varepsilon)\rho_r c_{pr})\frac{\partial T}{\partial t} = -\nabla\cdot(\vec{q}) - \rho_f c_p \vec{U}\cdot\nabla T - \left( \frac{\partial ln\rho_f}{\partial lnT} \right)_p \left( \varepsilon\frac{\partial p}{\partial t} + \vec{U}\cdot\nabla p \right) + \frac{\mu_f}{k} \parallel \vec{U} \parallel^2 \tag{15}$$

which now finally is the equation 4 given in the main text. In addition, $\frac{\partial ln\rho_f}{\partial lnT} = \frac{T}{\rho_f}\frac{\partial \rho_f}{\partial T} = -T\alpha_f$, where $\alpha_f \equiv -\frac{1}{\rho_f}\frac{\partial \rho_f}{\partial T}$ is defined as fluid thermal expansivity which is used in the equation 7 in the main text.

**Table 1.** Definitions and units of variables

| Symbol | Definition | Unit |
| --- | --- | --- |
| $t$ | Time | $s$ |
| $\varepsilon$ | Porosity | 1 |
| $\vec{q}$ | Conductive heat flux | $W\ m^{-2}$ |
| $\vec{U}$ | Velocity | $m\ s^{-1}$ |
| $p$ | Pressure | $Pa$ |
| $T$ | Temperature | $K$ |
| *Fluid and rock properties* | | |
| $\rho$ | Density | $kg\ m^{-3}$ |
| $h$ | Specific enthalpy | $J\ kg^{-1}$ |
| $u$ | Specific internal energy | $J\ kg^{-1}$ |
| $V$ | Specific volume | $m^3\ kg^{-1}$ |
| $c_p$ | Specific heat for a constant pressure process | $J\ kg^{-1}\ K^{-1}$ |
| $c_v$ | Specific heat for a constant volume process | $J\ kg^{-1}\ K^{-1}$ |
| $\alpha$ | Thermal expansivity | $K^{-1}$ |
| Subscript $f$ and $r$ denote fluid and solid(rock), respectively | | |

**References**

Bird, R. B., Warren E. S. and Edwin N. L.: Transport Phenomena, 2nd ed. Chapter 9-11. John Wiley & Sons, Inc. ISBN:0-471-41077-2

Feynman, Richard P., Robert B. Leighton, and Matthew Sands. The Feynman lectures on physics, Vol. I: The new millennium edition: mainly mechanics, radiation, and heat. Vol. 1., 2011. ISBN: 978-0-465-02562-6

---

## Author Comment (AC3) · 21 Sep 2020

We thank the reviewer for his time, precise summary, and positive evaluation of the manuscript.

Reply to reviewer's comments, all reviewer comments are in blue and our replies are in black.

1. l72-74: It is important to mention other initiatives that use OpenFOAM to solve flow and transport in porous media with Darcy-like solvers, e.g. Horgue et al. 2015 and Orgogozo et al. 2015

Yes, thanks for pointing that out - we have added the corresponding references to the manuscript.

2. l84: I think it is important here to mention that the PDEs are solved implicitly but sequentially.

   We have rephrased the sentence to mention the implicit and sequential scheme. The line 84 is rephrased as " *All the partial differential equations are solved implicitly in the framework of OpenFOAM and in a sequential scheme, and the thermal-physical models are developed using a pure water EOS.*"

3. l96: strictly speaking, in fluid mechanics, in laminar flow regime, there is inertia. Darcy's law corresponds to the creeping flow regime when inertia is negligible compared with viscous forces.

   Good point. We have clarified that and now refer to creeping flow in the main text.

4. Eq (2) : the right-hand side (=0) is missing

   We have fixed the issue in Eq (2).

5. l104: The compressibility of the rock is neglected as soon as the porosity is removed from the time derivative in Eq (1). Then, it is probably better to mention this hypothesis just after Eq (1).

   Yes, that's, of course, correct. The model is for constant-in-time porosity, i.e. incompressible grains. We have clarified that right after Eqn. (1).

6. the heat conductivity is noted lamba_r in Eqs (4)-(7) but k_r in the code.

   We have modified the $\lambda_r$ in Eqs (4)-(7) and table 1 to $k_r$ which keeps consistent with the code.

7. I am not sure that the energy equation in Eq (4) (and then Eq (7)) is exact. It seems correct only if the continuity equation is a divergence-free equation which

is not the case in the paper (this is what appears if you derive the equation from the conservation of the internal energy ddt(e) + div(e U) = div(q) + S with e(p,T) ). Moreover, it is not clear whether the two last terms of the right-hand side are necessary as they seem orders of magnitude lower than the other terms. Can you comment on that? Give an estimation of the weight of these terms and explain in which cases it is important to consider them?

As both reviewers had concerns about the assumptions and validity range of the energy equation, we have posted the full derivation as a separate author comment. We hope that that derivation is sufficiently clear to resolve questions about which terms are inside and outside the spatial and temporal derivatives.

That derivation is starting from the equation of change of internal energy, just as the reviewer pointed out. We have kept the pV and dissipation terms in the starting equation, both for completeness and because they actually matter for some submarine settings. But please see the full derivation for details.

Concerning the pV and dissipation terms (the last two terms in the energy equation): We did not find very many published studies but there is a nice pioneering paper by Garg and Pritchett ("On pressure-work, viscous dissipation and the energy balance relation for geothermal reservoirs", Advances in Water Resources, 1977). They show for geothermal systems that one either should keep both terms or none of them as they point to opposing directions. They also point out that for highly compressible cases (pure vapor, or supercritical fluid close to the critical point) they actually matter. We have made our own tests and find that also for large (say >5km) vertical extents (e.g. fault controlled systems at slow spreading ridges), these terms also matter.

That said, it is important to point out that the $-\left(\frac{\partial ln\rho_f}{\partial lnT}\right)_p \left(\varepsilon\frac{\partial p}{\partial t} + \vec{U} \cdot \nabla p\right)$ term is not exactly equal to pressure volume work but also contains the pressure dependence of enthalpy (see derivation). That terms again matters, for example, when

simulating fluid flow close to the critical point. Benchmarks C,D, where a fluid of constant temperature is flowing along a pressure gradient, show a drop in temperature on the lhs of the domain, which is partly related to this effect.

We have added a reference to the Garg and Pritchett paper in the main text.

8. Assuming that the equation is correct, the numerical treatment presented in Listing 10 can be improved by making the last term implicit with fvm::Sp( alphaP*(...) , T), which should lead to better numerical stability.

   Yes we agree. We have changed the last term of temperature equation to `fvm::Sp(alphaP*(porosity*fvc::ddt(p)+(U&fvc::grad(p))),T)` in the source code and Listing 1 in the manuscript.

9. l139: "adapt" instead of "adopt"

   Corrected.

10. l159-160: the sentence "since it is not straightforward to impose fluid velocities on boundaries" is not clear. Actually, here you just transport a velocity value into a boundary condition on the pressure gradient because your solver solves a pressure equation only.

    The sentence has been revised to "A similar boundary condition called `hydrothermalMassFluxPressure` is defined for the pressure field to prescribe a mass flux into the modeling domain."

11. l166: what is the difference between submarinePressure and OpenFOAM's PrghPressure?

    `submarinePressure` is a kind of fixed value boundary condition for pressure `p` , and it is specially designed for seafloor boundary of submarine hydrothermal problem. It provides hydrostatic pressure according to bathymetry or coordinate of seafloor patch. If user want use `submarinePressure` to set seafloor pressure boundary condition, the `y` coordinate must be negative because assuming

y of sea level is zero. However, prghPressure is designed for p_rgh . The submarinePressure is much more straightforward and easier to use for submarine hydrothermal modeling. The detail difference between them are shown below,

- *prghPressure*

  This boundary condition provides static pressure condition for p_rgh, calculated as:

$$p_{rgh} = p - \rho g(h - hRef) \tag{1}$$

  where

  - $p_{rgh}$: Pseudo hydrostatic pressure [Pa]
  - $p$: Static pressure [Pa]
  - $h$: Height in the opposite direction to gravity
  - $hRef$: Reference height in the opposite direction to gravity
  - $\rho$: density
  - $g$: acceleration due to gravity [m/s2]

  Code snippet

```
this->operator==
(
    *this - rhop*((g.value() & this->patch().Cf())
    - ghRef.value())
);
```

- *submarinePressure*

This boundary condition provides fixed hydrostatic pressure condition for `p` at **seafloor** boundary patch, which is derived from *fixedValueFvPatchScalarField* and calculated as:

$$p = p_a + \rho_{sw} * (\vec{g} \cdot \vec{C}_{patch}) \tag{2}$$

where

- $p$: pressure [Pa]
- $p_a$: pressure at sea level (or y=0 level), default is atmospheric pressure $10^5$ [Pa].
- $\rho_{sw}$: seawater density, default is 1013 [kg/m3].
- $\vec{g}$: gravitational acceleration vector [m/s2]
- $\vec{C}_{patch}$: face centres (coordinate) of seafloor patch [m]. **Note** that $y$ coordinate of seafloor surface must be negative when using `submarinePressure` boundary condition.

Code snippet

```
operator==
(
    p_Atmospheric + rhoValue_ * (g.value() & patch().Cf())
);
```

Example usage

```
seafloor
{
    type    submarinePressure;
    rhoValue 1013;  //Optional
}
```
12. l177: more details will be appreciated on the implementation of the thermo-physical model.

The thermo-physical model is implemented following the general guidelines for a thermo-physical model in OpenFOAM, e.g., `heHydroThermo` is modified from `$FOAM_SRC/thermophysicalModels/basic/heThermo`. Then just call thermal dynamic property calculation function of freesteam library, e.g. `freesteam_rho`, in the implementation function of the thermo-physical model, e.g. `species/water/equationOfState/IAPWSEOSI.H`. More details can be found in the source code.

We would like to leave it at that level of detail as a more in-depth discussion would distract from the main points of the paper.

13. l194-1999: the sentence about constrainPressure and fixedFluxPressure is very confusing. Why do you need fixedFluxPressure in your simulations? What is the link with the boundary conditions introduced in Section 2.4?

Actually the `fixedFluxPressure` boundary condition and the `hydrothermalMassFluxPressure` (introduced in section 2.4) are similar, the first one is the OpenFOAM's internal BC which requires `U`, while the second one doesn't need `U`. Although `fixedFluxPressure` boundary condition is not used frequently in our simulation, in order to make the solver compatible with this boundary condition, the function of `constrainPressure` has to be added to update pressure boundary condition before solving `p` in the pEqn.C file. In addition, `constrainPressure` only works when pressure boundary condition is `fixedFluxPressure`, otherwise it doesn't do anything.

14. It is difficult for the potential user to know where is the code. It is scattered on too many platforms. The document mentions at least 3 different locations (Zenodo.org, DockerHub, GitLab). In particular, Zenodo.org and GitLab seem to provide the same code.

Thanks for these comment, indeed there are two many locations, we have add necessary explanations to each repositories in the code availability section.

- GitLab repository always contains the latest development version of the source code.
- Zenodo repository is selected only for the GMD publication because it provides a DOI number. It will be updated in the future if we have new release version of HydrothermalFOAM.
- Docker Hub repository contains all the runtime environment, e.g. Open-FOAM, Gmsh, python, HYDROTHERM, etc., required by Hydrothermal-FOAM and benchmark examples in the manuscript. This repository will not be updated in the future unless some new runtime environment is required by HydrothermalFOAM.

Therefore, the potential user should only follow the GitLab repository for the source code. The Docker Hub version is for the new users who are not familiar with OpenFOAM or doesn't have OpenFOAM.

We also post an author comment of How to update HydrothermalFoam to describe how fetch and use the latest source code.

15. l354-355, l364-365, l374-375: "can be be", "bechmarks"

We have corrected all these typo issues in the manuscript.

16. l396 Zenodo.org

We have corrected this typo in the manuscript.

17. Section 5: I think it will be interested to have a comparison of the simulation time of HYDROTHERM and HydrothermalFoam. Such a comparison can highlight better the importance of using modern computational platforms.

We have compared the simulation time of HYDROTHERM and Hydrothermal-FOAM for 3D benchmark examples described in the manuscript (section 3.2). HydrothermalFOAM is ∼24 times faster than HYDROTHERM in serial computing case. In addition, HydrothermalFOAM has parallel computing ability but HYDROTHERM seems not. Therefore, maybe it is not necessary to compare the simulation time of them. While the simulation time of these two solvers for the 3D benchmark examples are listed in table 1 to answer your question.

We'd prefer to not enter that discussion in the main text because it is not an entirely fair comparison and because better performance with respect to HYDROTHERM was not our motivation to develop the Openfoam-based model.

**Table 1.** Simulation time comparison of HYDROTHERM and HydrothermalFOAM for 3D benchmark models

| Simulator | 3D Homogeneous model | | 3D Heterogeneous model | |
|---|---|---|---|---|
| | Serial | Parallel (4 cores) | Serial | Parallel (4 cores) |
| HydrothermalFOAM (h) | 1.2575 | 0.4872 | 0.8081 | 0.5386 |
| HYDROTHERM (h) | 25.0403 | - | 30.2556 | - |

---

## Author Comment (AC4) · 21 Sep 2020

We thank the reviewer for his time, precise summary, and positive evaluation of the manuscript.

Reply to reviewer's comments, all reviewer comments are in blue and our replies are in black.

1. Table 1, definition of the time step: If possible, write times in lowercase, otherwise they can be confused with temperature.

   Thanks, we have change $\Delta T$ to $\Delta t$ in Table 1.

[Figure]

2. Eq. (2): This is not an equation. "= 0" is missing

   Corrected.

3. Eq. (4): This equation is correct even in the compressible regime but only if the rock porosity, density and heat capacity are constant in time, and if the fluid heat capacity is constant both in space and time. These assumptions should be stated explicitly in the text. However, I am afraid that the assumptions on the fluid heat capacity are not fulfilled in this paper, because $Cp\_f$ is treated as a VolScalarField, and also because a detailed thermodynamics for water is used (Figure 1, L113 and Listing 4). In this Lagrangian formulation, to correclty consider the variability of the fluid heat capacity, a term proportional to its partial time derivative and another proportional to its gradient should be added.

   Also the the co-reviewer Cyprien Soulaine had concerns about the energy equation. We have therefore posted a detailed derivation as an author comment, which we hope resolves these questions.

   In short: yes, the solid matrix is incompressible and has constant properties. The fluid properties are determined from the EOS of pure water and vary with pressure and temperature. The reason why density and specific heat end up outside the derivatives is not because they are assumed constant, but because of the thermodynamic identities between enthalpy and temperature - and the use of the mass conservation equation. We hope that the derivation will clarify these questions!

4. Given a certain resolution, the typical time scale for which this approximation holds should be given. For example, using the data in Table 1, the given rock thermal diffusivity is $lambda\_r/(Cp\_r*rho\_r) \sim$ 6e-7 m2/s. This means that, roughly, the typical thermal equilibrium time for a resolution of 100 m is order of 500 years. Thus, for the approximation to hold, the typical time of the dynamics at this resolution should be larger than, roughly, 500 years. In other words, the equilibrium

time is related to the natural time step used by the simulation. The regime where this equilibrium assumption holds should be given, to allow the user to use the code in its proper regime of approximations. For an example on this issue, see Remark 2.1 in: Cerminara, M., & Fasano, A. (2012). Modeling the dynamics of a geothermal reservoir fed by gravity driven flow through overstanding saturated rocks. Journal of Volcanology and Geother- mal Research, 233–234, 37–54. https://doi.org/10.1016/j.jvolgeores.2012.03.005

Cool paper and elegant way of investigating the equilibration time scales! However, we think this issue is resolved by the derivation of the energy equation.

5. L123: The description is not following the terms order in Eq. (5). Please correct this.

The terms order of Eq. (5) has been revised to be consistent with the text.

6. Eq. (6): This reformulation iterates the problem with the spatial variations of the fluid heat capacity, see comment above on Eq. (4) and below on Eq. (7).

This all goes back to how the energy equation is written. The way we did it, and which we feel is correct, requires these mathematical reformulations in order to implement the equation in Finite Volumes, which are great for solving divergence terms but not that much for velocity times gradient terms.

It can be debated if a scheme that takes specific enthalpy or internal energy as a primary variable instead of temperature would have been a better choice. In an energy-based scheme, we could have kept the energy equation in clean conservative/divergence form, which is better for FV solutions. We chose temperature as primary variable because it is so intuitive for the user but the downside is that it requires some re-formulation of the energy equation.

7. Eq. (7): The term proportional to the divergence of (rho_f*Cp_f*U) should not be present when the spatial dependence of the fluid heat capacity is correctly taken

into account. A term for its temporal dependence is still missing.

Please see derivation as spelled out in the author comment.

8. L171: Here should be clearly stated that the EOS is used in the single phase regime, no phase transition is admitted in this formulation.

Yes, we should have made that clear - it's fixed now!

9. L119 (should be L219): Usually in OpenFOAM, spatial schemes are first or second order, not high order. Using the upwind interpolation makes the code first order in space. Some word on the time scheme used is also needed.

We have rephrased the sentence to "In the following benchmark tests (section 5), the advective discretization scheme is set to upwind to ensure consistency with HYDROTHERM. It should be noted that all of the basic numerical schemes of OpenFOAM are also valid for *HydrothermalFOAM* solver."

---

## Author Comment (AC5) · 19 Oct 2020

**1   Physical meaning of terms in the equation of change of internal energy**

As the discussion of the paper is still open, we would like to take the chance to elaborate a bit more on the importance and meaning of the different terms in the equation of change of internal energy. We had posted previously how that equation can be written with temperature as primary variable. We hope that the underlying thermodynamic identifies and mathematical transformations are sufficiently clear now and that the presented benchmarks in the main text convincingly demonstrate their correctness.

[Figure]

The resulting "temperature" equation contains two terms that are often neglected and the reviewers encouraged us to elaborate more on those terms. We did so in the revised version of the manuscript but did not provide an in-depth discussion. The reason being that we would like to have the main text focused on making a state-of-the-art and well-documented 3-D hydrothermal flow model available to the wider submarine hydrothermal community. That is what this paper is about.

That said, we also feel that GMD's "author comments" provide the space to have these extra discussions. So let's discuss those terms! This is the equation of change of internal energy written in terms of temperature as primary variable. It's detailed derivation can be found in the accompanying author comment #2. The terms in question are marked in purple.

$$
\begin{aligned}
(\varepsilon \rho_f C_{pf} + (1-\varepsilon)\rho_r C_{pr})\frac{\partial T}{\partial t} = \nabla \cdot (k_r \nabla T) - \rho_f C_{pf}\vec{U} \cdot \nabla T \\
+ \frac{\mu_f}{k} \parallel \vec{U} \parallel^2 - \left(\frac{\partial ln\rho_f}{\partial lnT}\right)_p \left(\varepsilon \frac{\partial p}{\partial t} + \vec{U} \cdot \nabla p\right)
\end{aligned}
\tag{1}
$$

The first term describes viscous dissipation, so has a clear physical meaning. A discussion of it can be found in Garg and Pritchett (1977). The second term contains two components: a component of pressure volume work that is expressed as the substantial derivative of pressure, when reformulating the equation of change of internal energy in terms of enthalpy (compare equations 1 & 8 in Author Comment #2) and, more importantly, the pressure dependence of enthalpy (Fig. 1). Remember that we switch from enthalpy to temperature as primary variable by using the thermodynamic identity:

$$
dh_f = \frac{\partial h_f}{\partial T}_p dT + \left(\frac{\partial h_f}{\partial p}\right)_T dp = c_p dT + \left[V - T\left(\frac{\partial V}{\partial T}\right)_p\right] dp
\tag{2}
$$

While the specific heat, $c_p$, describes how specific enthalpy changes with temperature, the term in square brackets describes how it changes with pressure. Fig. 5 shows $\left(\frac{\partial h_f}{\partial T}\right)_p$ and $\left(\frac{\partial h_f}{\partial p}\right)_T$ as functions of $p$ and $T$ for pure water. In the incompressible regime, enthalpy is mainly a function of temperature; under near critical, and vapor/vapor-like conditions when the fluid becomes increasingly compressible, the pressure-dependence of enthalpy starts to matter.

The physical meaning of these terms in equation 1 becomes clear when we consider the identity $\frac{\partial ln\rho_f}{\partial lnT} = \frac{T}{\rho_f}\frac{\partial \rho_f}{\partial T} = -T\alpha_f$, where $\alpha_f \equiv -\frac{1}{\rho_f}\frac{\partial \rho_f}{\partial T}$. With this, we can write the energy equation as:

$$(\varepsilon\rho_f C_{pf} + (1-\varepsilon)\rho_r C_{pr})\frac{\partial T}{\partial t} = \nabla\cdot(k_r\nabla T) - \rho_f C_{pf}\vec{U}\cdot\nabla T$$
$$+\frac{\mu_f}{k}\parallel\vec{U}\parallel^2 +T\alpha_f\left(\varepsilon\frac{\partial p}{\partial t} + \vec{U}\cdot\nabla p\right) \qquad (3)$$

The last term now takes a more familiar form, it describes adiabatic effects.

To visualize this, we go back to the three 1-D benchmark tests presented in the main text but now we also present the results, when the "purple" terms in the energy equation are neglected (dashed purple lines). The first test is in the liquid single-phase regime, the fluid is largely incompressible, and the terms in question do not matter (Fig. 2). All solutions match the reference solution obtained with HYDROTHERM (Fig. 2). The situation is different for the second, near critical, and third, pure vapor, test (Fig. 3 and 4). Here the pressure dependence of enthalpy does matter and the reference solution can only be matched when it is accounted for. Remember that HYDROTHERM neglects viscous dissipation but solves the equation of change of internal energy in conservative divergence form with specific enthalpy as primary variable (Faust and

Mercer, 1979):

$$\frac{\partial(\varepsilon\rho_f h_f + (1-\varepsilon)\rho_r h_r))}{\partial t} = \nabla \cdot (k_r \nabla T) - \nabla \cdot (\rho_f h_f \vec{U}) - \frac{\partial \varepsilon p}{\partial t} + \vec{U} \cdot \nabla p \qquad (4)$$

The final question is if those terms matter for submarine hydrothermal systems. Here the answer obviously depends on the setup and especially the vertical extent of the circulation zone - but in general the answer is "yes". High temperature hydrothermal systems have been shown to operate close to maximum efficiency (Jupp and Schulz, 2000, 2004). A consequence is that the upwelling temperature is typically the one that maximizes heat transport by buoyant flow (Driesner 2010). Jupp and Schulz, (2000) developed the concept of "fluxibility", a property that describes how efficient buoyant heat transport is. Hydrothermal plumes tend to detach from a thermal boundary layer, where F is large. Fig. 6 shows that the region where $F$ is large corresponds to those $p, T$ conditions, where adiabatic effects matter. That is the reason why submarine hydrothermal flow models should account for that term. We hope that readers will find these extra thoughts and considerations helpful, but, for the reasons given above, we have not included them into the main text.

**Fig. 1.** Enthalpy as a function of pressure and temperature. Dashed lines are contours of enthalpy. The other annotations are the same as what in figure 1 in the manuscript.

**Fig. 2.** Flow along a pressure gradient in the single-phase liquid regime (Model 1). Enthalpy is only weakly dependent on pressure and all solutions match the reference HYDROTHERM solution. The purple lines are for a model that does not account for the "purple" adiabatic and dissipation terms in equation 1&3.

**Fig. 3.** Flow along a pressure gradient at near-critical conditions (Model 2). The temperature drop within the from the left invading hot fluid is related to adiabatic effects. Note in Fig. 1 (white line) how enthalpy is constant and temperature drops along this limb of the flowpath. The purple lines are for a model that does not account for the "purple" adiabatic and dissipation terms in equation 1&3.

**Fig. 4.** Flow along a pressure gradient at pure vapor conditions (Model 3). Again the pressure dependence of enthalpy matters and the reference solution is only matched, when the respective (purple) terms are present in the energy equation. The purple lines are for a model that does not account for the "purple" adiabatic and dissipation terms in equation 1&3.

**Fig. 5.** Enthalpy change with pressure and temperature. $dH/dp$ and $dH/dT$ as a function of pressure and temperature are shown in (a) and (b), respectively. The other annotations are the same as what in Fig. 1. Note that panel (a) shows absolute values.

**Fig. 6.** Adiabatic temperature gradient (a) and normalized fluxibility (b).

**References**

Garg, S. K., & Pritchett, J. W. (1977). On pressure-work, viscous dissipation and the energy balance relation for geothermal reservoirs. Advances in Water Resources, 1(1), 41–47. https://doi.org/10.1016/0309-1708(77)90007-0

Jupp, T., & Schultz, A. (2000). A thermodynamic explanation for black smoker temperatures. Nature, 403(6772), 880–883. https://doi.org/10.1038/35002552

Jupp, T. E., & Schultz, A. (2004). Physical balances in subseafloor hydrothermal convection cells. Journal of Geophysical Research: Solid Earth, 109(B5). https://doi.org/10.1029/2003JB002697

Driesner, T. (2010). The interplay of permeability and fluid properties as a first order control of heat transport, venting temperatures and venting salinities at mid-ocean ridge hydrothermal systems. Geofluids, 10(1-2), 132–141. https://doi.org/10.1111/j.1468-8123.2009.00273.x

Faust, C. R., & Mercer, J. W. (1979). Geothermal reservoir simulation: 1. Mathematical models for liquid- and vapor-dominated hydrothermal systems. Water Resources Research, 15(1), 23–30. https://doi.org/10.1029/WR015i001p00023

[Figure]

[Figure]

Fig. 1.

[Figure]

**Fig. 2.**

[Figure]

**Fig. 3.**

[Figure]

**Fig. 4.**

[Figure]

**Fig. 5.**

Fig. 6.

---

## Editor Comment (EC2) · Thomas Poulet (Editor) · 20 Oct 2020

Thank you very much for this extra information. This is is a good example of how to use the "authors comments" section of the discussion indeed, to focus in more details on a point that is important but that would have nonetheless detracted the manuscript from its main purpose. (Note that it also opens the door for the reviewers to suggest that some of this content gets added as appendix in the main text if need be.)

---

## Author Response (AR2)

We thank the reviewers for their time, precise summary, and positive evaluation of the manuscript.
Reply to reviewer's comments, all reviewer comments are in blue and our replies are in black.

**Replay to reviewer #3**

1. Line 33: The plural of progress is progress not 'progresses'. –> significant progress has been made ...

   'progresses' in both line 33 and line 37 have been corrected.

2. Line 76: ... hydrothermal circulation in 'a' single-phase regime ..

   It has been corrected.

3. Line 81: be consistent with singular/plural choice: "structured meshes, unstructured mesh, mixed mesh"

   Fixed.

4. Equation (2): I was at first confused that there is no porosity in the second term of the equation as well (experts in hydrothermal flow may be more familiar with the current form, but I am more used to other applications). I have traced this form of the equation and found the same formulation (without the porosity and without explanation) in Hasenclever et al., 2014, which further refers to Theissen-Krah, S., Iyer, K., Rüpke, L. H., & Morgan, J. P. (2011), which finally provides an explanation on the assumptions that allow this simplification (equation (8) and (9) in that paper). I would suggest to put a reference here to point to the original derivation, to make this process simpler.

   Thanks! We have added this reference.

5. Equation (8): I could not find a definition of Phi for this equation, and the index i does not appear either (I suppose it is meant to be F, the index of the sum?).

   Yes, you are right. $F$ is the index of the sum and $i$ should be the face index. Therefore we have corrected them to be consistent. We have added definition of $\phi_F$, $V_{cell}$ and $S_f$ in equation (8).

   $$Co = \max_{\forall cell} \left( \frac{\sum_{F=1}^{m} \phi_F}{2\rho_f V_{cell}} \right) \Delta t$$

   with $m$ being the number of neighbor faces to a specific cell, $\phi_F = \rho_f \boldsymbol{U} \cdot \boldsymbol{S_F}$ being mass flow through the cell faces, $V_{cell}$ being volume of the cell and $\boldsymbol{S_f}$ being area of face $F$, respectively.

6. Equation (9): Co_fixed is not defined either.

   $Co_{fixed}$ is the maximum Courant number specified in `system/controlDict` file of a case. We have modified $Co_{fixed}$ to $Co_{max}$ to keep consistent with the source code and also added symbol definition in Table 1 in the main text.

7. Equation (10): This equations seems to only limit the growth of the timestep, is there no reason to limit reductions of time step length?

   Reductions of time step length is controlled by maximum Courant number $Co_{max}$ as well. For example, in the early simulation stage, $Co$ should be small because velocity is small, therefore $C_{\Delta t}$ could be greater than 1, $\Delta t$ will increase but limits to a maximum increase of 20%. However, if velocity becomes bigger thus $Co$ becomes bigger, then $C_{\Delta t}$ could be less than 1 and the time step length $\Delta t$ will decrease according to equation (10). In addition, a maximum time step length ($\Delta t_{max}$) can also be specified by keyword `maxDeltaT` in `system/controlDict` file, which is used to limit the global time step length using equation of $\Delta t = min(min(min(C_{\Delta t}, 1 + 0.1C_{\Delta t}), 1.2)\Delta t_{last}, \Delta t_{max})$.

   Of course the proper $Co_{max}$ depends on the specific modeling problem and specific solver, generally speaking $Co_{max} = 0.8$ works well for most of cases for `HydrothermalSinglePhaseDarcyFoam` solver.

8. Line 161: You may want to mark the keyword as italic or bold or put it in quotation marks to make the importance of it more clear.

   We have marked all the keywords using LaTeX command of `\texttt{}`.

9. Line 170: ... "for the" pressure field for ...

   It has been corrected.

10. Line 171: ... for "the" pressure field ...

    It has been corrected.

11. Line 191: ... very "close" to the mathematical formulation

    It has been corrected.

12. Line 215: ... propertie of fluid ... -> properties of the fluid

    It has been corrected.

13. Section 4: I would advise the authors to read through this section and make sure all terminal commands, file names, directory names, and input file keywords are marked clearly as such, e.g. by using the Latex ' texttt{}' command or using an appropriate font in Word/LibreOffice. Currently many commands are hard to distinguish from surrounding text, e.g. Line 265, 268, 269, 271, 278, 280, 300, 301, 302, 306 and others. Section 4 would also benefit from a careful proof-reading as some sentences seem to miss a number of articles.

    Thanks for the advice, we have marked all commands, keywords, file names and directory names in the main text by using LaTeX `\texttt{}` command.

14. Lines 302-304: This sentence is not clear to me, why has the 'boundaryField' key to be set? Please extend the explanation somewhat (maybe split the sentence in two).

    The sentence has been rephrased as "While a boundary condition for permeability is not mathematically required, its internal OpenFOAM datatype requires a corresponding `boundaryField` dictionary with specified boundary conditions. Therefore we suggest that the `type` entry of all boundary patches for permeability field should be always set to `zeroGradient`."

15. Figure 7: In the caption: HYDROTHER -> HYDROTHERM. ... The visualization window are shown by black box ... -> The visualization window "is" shown by "the" black box ...

    It has been corrected.

16. Line 390: meshed -> meshes

    It has been corrected.

**Reply to reviewer #2**

1. The paper in the present form explains in a clearer way the hypothesis behind the model. Thanks to the author for the derivation of the energy equation. Now I agree that, given a porosity that is constant in time, the energy equation is correct, in its formulation with respect to both specific enthalpy and temperature. At my advice, the only point that is still missing is that regarding the typical time scale for which the local thermal equilibrium holds. The energy equation is correct under this assumption, that holds only if the dynamics is slow enough. I think it would be useful for a reader to know the minimum time scale of the dynamics enabling the local equilibrium assumption.

   Thanks, we should have made that clear during the last iteration! We have now added the sentences below to the main text so that the reader is well aware of the thermal equilibrium assumption and under which special conditions this assumption

might not hold anymore. We feel that a more detailed analysis of the equilibrium time scales is beyond the scope of the paper.

"...Note that the assumption of thermal equilibrium is valid for most practical applications in submarine hydrothermal system modeling, as the equilibration time scale is short being related to grain size in a porous medium. However, this assumption should be carefully reviewed when simulating special geometries like fluid-filled cracks (SchmelingGJI2018) or very short time scales like the response of a hydrothermal circulation cell to a seismic event (Wilcock2004). For such specialized cases OpenFOAM offers support for multi-physics models that can resolve different physics in differing parts of the modeling domain, so that heat transfer between solid and fluid can be explicitly resolved. Under the equilibrium assumption, changes..."

**Reply to reviewer #1**

1. I am satisfied with the revised version and author's comments.

**Annotation of changes**

- Added: new added text

- Replace text: new text

- Replace math: $\alpha_{new}\underline{\beta_{old}}$

- Delete:

**List of changes**

[revised manuscript text omitted]